



# On the estimation of boundary layer heights: A machine learning approach

Raghavendra Krishnamurthy[1], Rob K. Newsom[1], Larry K. Berg[1], Heng Xiao[1], Po-Lun Ma[1], David D. Turner[2]

[1]Pacific Northwest National Laboratory, ASGC Division, Richland, 99354, USA
[2]National Oceanic and Atmospheric Administration / Global Systems Laboratory, Boulder, 80305, USA

*Correspondence to*: Raghavendra Krishnamurthy (raghu@pnnl.gov)

**Abstract.** The planetary boundary-layer height ($z_i$) is a key parameter used in atmospheric models for estimating the exchange of heat, momentum and moisture between the surface and the free troposphere. Near-surface atmospheric and subsurface properties (such as soil temperature, relative humidity etc.) are known to have an impact on $z_i$. Nevertheless, precise relationships between these surface properties and $z_i$ are less well known and not easily discernable from the long-term data. Machine learning approaches, such as Random Forest (RF), which use a multi-regression framework, help to decipher some of the physical processes linking surface-based characteristics to $z_i$. In this study, a multi-year dataset from 2016 to 2019 at the Southern Great Plains site is used to develop and test a machine learning framework for estimating $z_i$. Parameters derived from Doppler lidars are used in combination with over 20 different surface meteorological measurements as inputs to a RF model. The model is trained using radiosonde-derived $z_i$ values spanning the period from 2016 through 2018, and then evaluated using data from 2019. Results from 2019 showed significantly better agreement with the radiosonde compared to estimates derived from a thresholding technique using Doppler lidars. Noteworthy improvements in daytime $z_i$ estimates was observed using the RF model, where a 50% improvement in mean absolute error compared to lidar-only $z_i$ estimates and provided an $R^2$ of greater than 85%. We also explore the effect of $z_i$ uncertainty on convective velocity scaling and present preliminary comparisons between the RF model and $z_i$ estimates derived from atmospheric models.

## 1 Introduction

Measuring the growth of the Planetary Boundary Layer (PBL) height is crucial for understanding the turbulent transfer of air mass, which in turn strongly influences the winds, temperature, and moisture within the atmospheric boundary layer. During daytime, the air within the PBL is well mixed due to convection and results in weakening of turbulence at the top of the PBL (entrainment zone). One of the characteristics of the top of the PBL is that the turbulence is near zero. Routine, continuous, long-term monitoring of the PBL height, $z_i$, is crucial for evaluating climate, weather, and air quality models in



representing near-surface turbulent mixing, entrainment across the PBL top, the development of shallow cumulus, understand

effects of morning or evening transition (Grant 1997), and nocturnal convection initiation (Reif et al., 2017). Profiles of

potential temperature, water vapor mixing ratio, and particulate concentration often exhibit strong gradients at or near the top

of the PBL. Typically, estimates of $z_i$ are obtained from an analysis of temperature and humidity profiles obtained from

radiosondes. Indeed, radiosondes continue to be the defacto standard due to their long operational history, accuracy, and

reliability. However, major limitations are the poor temporal resolution and large sampling error since radiosondes are typically

only launched twice daily at operational centers around the world.  The launch time periods are generally not optimal for

looking at various boundary layer properties.  As a result, the diurnal variation in $z_i$ is usually poorly represented in radiosonde

data.

Modern remote sensing instruments can provide continuous estimates of the boundary layer dynamics.  Space-based

remote sensing instruments, such as Moderate Resolution Imaging Spectroradiometer (MODIS), Tropical Rainfall Measuring

Mission (TRMM) data, and Multiangle Imaging SpectroRadiometer (MIRS) have also shown the ability to estimate PBL

height (Wood and Bretherton 2004, Karlsson et al. 2010).  Since these measurements are not continuous, a thorough evaluation

of such techniques has not yet been conducted due to limited data.

Ground-based lidar systems, such as the Raman lidars (Turner et al., 2014), ceilometer, micro pulse lidar (Campbell

et al. 2002), atmospheric emitted radiance interferometer (Knuteson et al. 2004; Sawyer and Li 2013), Doppler lidar (Tucker

et al., 2009, Berg et al., 2017) and high spectral resolution lidar have been commonly used to provide PBL height estimates

(Quan et al., 2013, McNicholas & Turner 2014).  For elastic lidar systems such as the micro pulse lidar, $z_i$ is estimated by

locating the height where the range-corrected Signal to Noise Ratio (SNR) or attenuated backscatter profile experiences a

strong decrease with height (Emeis et al., 2008). A similar approach is used to estimate $z_i$ from profiles of water vapor mixing

ratio from Raman lidar (Summa et al., 2013) or differential absorption lidar systems (Hennemuth & Lammert 2006 ); however,

the peak in the water vapor variance has also been used as an estimate of $z_i$ (Turner et al. 2014).  Alternatively, velocity

information from a Doppler lidar can be used to estimate $zi$.

Doppler lidars provide range-resolved measurements of radial velocity, attenuated backscatter, and signal to noise

ratio (SNR).  When staring vertically, a ground-based Doppler lidar measures height-resolved profiles of vertical velocity in

the lower atmosphere with a temporal resolution of 1 second or less. Profiles of the vertical velocity variance can then be

computed by averaging over an appropriate time interval (typically 15 to 30 minutes).  The primary advantage of Doppler

lidars is that they measure the turbulence directly and thus provide a more defensible measure of the boundary-layer depth.

Other systems need to relay on gradients of aerosol loading or moisture that are used to infer the boundary-layer depth.

One method for estimating the Convective Boundary Layer (CBL) depth is to find the height where the vertical

velocity variance profile falls below some prescribed threshold, which in some cases can vary with time (Lenschow et al.,

2000, Tucker et al., 2009, Lenschow et al., 2012, Berg et al., 2017). This method, which we refer to as the Tucker method, is

simple to implement and provides a direct measure of the mixed layer depth. However, the estimates are sensitive to the choice

of variance threshold, which is somewhat arbitrary.  Also, this method fails under stable nocturnal conditions due to weak



turbulence and the fact that the minimum range of the lidar is often above the depth of the nocturnal $z_i$. Although, one can

estimate the PBL height based on wind shear and Turbulence Kinetic Energy (TKE), there has been limited research on this topic (Brost and Wyngaard 1978, Teixeira and Cheinet 2004, Le Mone et al., 2018).

Due to different measurement approaches between multiple remote sensing instruments, considerable uncertainties exist when comparing it to standard radiosonde retrievals. PBL heights from different instruments provide expected trends during certain atmospheric conditions (mostly daytime convective time periods) but differ slightly due to measurement

uncertainties and thresholds chosen associated with each instrument. Therefore, a framework independent of threshold techniques used in previous studies is warranted. Although this paper does not directly address a unified approach to estimate $z_i$, it is a step in that direction.

In view of these limitations, we investigate the potential of using a Machine Learning (ML) approach for continuous monitoring of $z_i$, with a focus on the CBL. ML enables us to bring together various observations in order to come up with a

consensus answer. ML models, such as RF and Neural Networks, have been used for classifying various atmospheric phenomenon (McGovern et al. 2017, Gagne II et al., 2019, Vassalo et al., 2020) or retrieving atmospheric variables (e.g., Solheim 1998, Cadeddu et al. 2009, Bianco and Wilczack 2002, Bonin et al., 2018). The RF model is versatile, simple to implement, and robust. The training of the RF model entails providing it with observations (i.e. features) which aide in predicting $z_i$. Examples of such observations include surface sensible and latent heat flux, soil temperature, soil moisture,

surface potential temperature, surface humidity etc. These features have all been shown to exhibit some degree of correlation with $z_i$ (Santanello et al., 2005; Zhang et al., 2013).

Here we use a RF model to predict the boundary layer height using input features derived from vertically-staring Doppler lidar data and various surface and sub-surface observations. We use a multi-year dataset from the US Department of Energy's (DOE) Southern Great Plains (SGP) site (Sisterson et al. 2016) for training and evaluating the model. Reference $z_i$

measurements from radiosondes are used in the RF training process (Sivaraman et al., 2013). Specifically, the RF model is trained using observations from 2016 through 2018, and its performance is evaluated using data from 2019.

In this paper, Section 2 describes the observations that are used in this study, Section 3 describes the details of the RF model, including the training method, data conditioning and performance. Results of the RF model's performance are presented in Section 4, and in Section 5 we examine how RF model-derived $z_i$ estimates impact the scaling of the vertical velocity

variance profiles. PBL heights estimates from Energy Exascale Earth System Model (E3SM) Atmosphere Model version 1 (EAMv1), large-eddy simulations (LES), and data are compared in Section 6. Finally, summary and future work is provided in Section 7.



## 2 Data Sources

The US DOE Atmospheric Radiation Measurement (ARM) User Facility operates the Southern Great Plains (SGP) site in north-central Oklahoma (Mather and Voyles 2013; Sisterson et al 2016). The site contains an extensive suite of instrumentation for monitoring the atmosphere and surface properties. Most of these instruments operate continuously and the data are freely available from the ARM website (https://adc.arm.gov/discovery /; McCord and Voyles 2016). The SGP site contains several heavily instrumented subsites or "facilities" that are geographically dispersed over Oklahoma and Kansas (Mather and Voyles

2013). For this study, we use observations from the central facility (C1), which also contains the largest number and most diverse suite of instruments. Figure 1 shows the layout of the SGP C1 site and the locations of instruments used in this study. Additionally, Table 1 lists the instruments, ARM data stream names, and specific measurements that were used. In ML, independent variables, or the inputs, are often referred to as features. A model can have multiple features/inputs and for this project, the measurements from the observations will be referred to as features in the RF model.


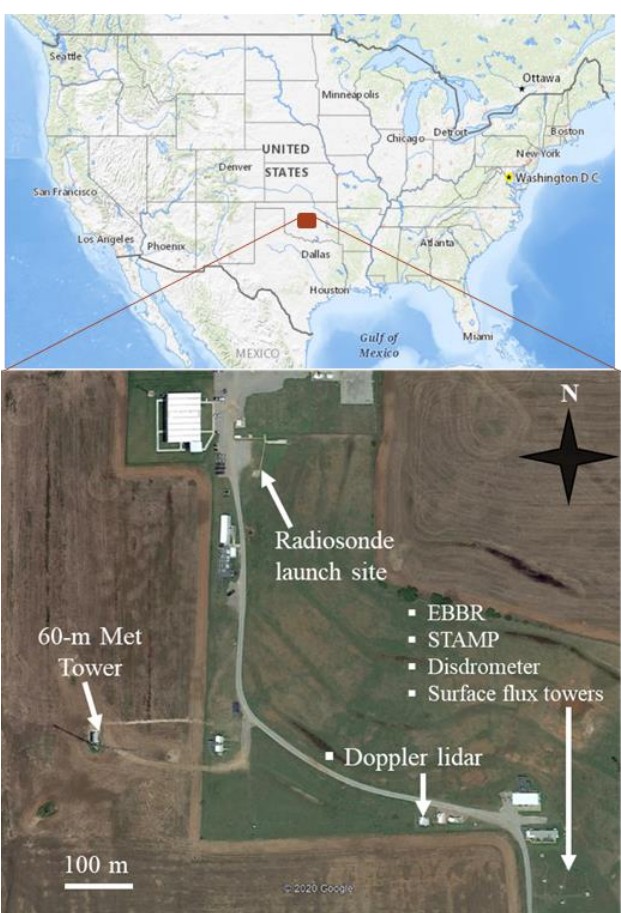

**Figure 1. ARM SGP site central facility (C1) layout and instruments used in this study in Oklahoma, USA. Maps are extracted from © Google Earth and © Google Maps.**





**Table 1: Instruments, ARM data stream names, and measurements used in this study.**

| Instrument | ARM Data stream | Measurements or Features | References |
|---|---|---|---|
| Radiosonde | sgppblhtsonde1mcfarlC1.c1 | PBL height estimates | Sivaraman et al., 2013 |
| Surface Eddy Correlation Station (ECOR) | sgp30ecorE14.b1 | Sensible heat flux | Cook et al., 2018 and Tang et al., 2019 |
| | | Latent heat flux | |
| | | vertical velocity variance | |
| | | Friction velocity | |
| | | Turbulence Kinetic Energy | |
| | | Monin-Obukhov length | |
| | | Surface wind speed | |
| | | Surface wind direction | |
| | | Momentum flux | |
| Surface Meteorological Station (MET) | sgpmetE13.b1 | Air Temperature | Ritsche and Prell 2011 |
| | | Relative humidity at 4m | |
| The Soil Temperature And Moisture Probes | sgpstampE14.b1 | Soil moisture at -5cm Soil temperature at -5cm | Cook 2018 |
| surface energy balance system / solar infrared radiation station | sgpbeflux1longC1.c1 | Best estimate of longwave, shortwave and normal radiation | Cook and Sullivan 2019 |
| Doppler lidar (DL) | sgpdlfptC1.b1 | Vertical velocity, range-corrected attenuated backscatter, signal to noise ratio (SNR) variance from surface to 800 m AGL | Newsom and Krishnamurthy 2020 |
| | sgpdlprofwstats4newsC1.c1 | Height-resolved vertical velocity variance, cloud base height | Newsom et al., 2019b |
| | sgpdlprofwinds4newsC1.c1 | Height-resolved wind speed and direction | Newsom et al., 2019a |
| | sgpdlfptC1.b1 | Average eddy dissipation rate from surface to 800 m AGL | Champagne et al., 1977 |
| | sgpdlprofwstats4newsC1.c1 | CBL depth from the Tucker method | Tucker et al., 2009 |
| | sgpdlprofwinds4newsC1.c1 | Wind shear exponent ($\alpha = \log_{10}\left(\frac{U_1}{U_2}\right)/\log_{10}\left(\frac{Z_1}{Z_2}\right)$), where $U_i$ and $Z_i$ are wind speed and height at altitude $i$. | Wharton and Lundquist 2012. |



The ARM User Facility operates a Doppler lidar at SGP C1. The instrument is a Halo Photonics Stream Line XR (Pearson et al. 2009), and has operated at C1 since April 2011. The instrument provides height- and time-resolved measurements of radial velocity, attenuated backscatter and SNR. The range resolution is set to 30 m and the temporal

resolution is about 1-s. The instrument is configured to stare vertically most of the time. Once every 15 minutes it executes a plan-position-indicator scan, from which profiles of the wind speed and direction are computed. The vertical staring data are used to compute profiles of noise-corrected vertical velocity variance using a 30-minute averaging period. More details about the instrument are provided by Newsom and Krishnamurthy (2020). Details about the vertical velocity statistics value-added product (VAP) are provided by Newsom et al (2019a), and detail about the Doppler lidar wind VAP are given by Newsom et

al (2019b).

Doppler lidar-derived features used in this study are listed in Table 1. The list includes raw height-resolved measurements of attenuated backscatter and range-corrected SNR, which are known to be directly correlated with the boundary layer height (Cohn & Angevine 2000, Brooks 2003). Also listed are several derived quantities such as cloud base height, wind shear, turbulence eddy dissipation rate, and $z_i$ estimated using the Tucker method (Tucker et al., 2009). Estimates of eddy

dissipation rate were computed between 100 to 800 m AGL from the 1 Hz vertical velocity data using the method described by Champagne et al., 1977.

The ARM PBL VAP (sgppblhtsonde1mcfarlC1.c1) contains estimates of $z_i$ derived from radiosondes launched at SGP C1. We note that radiosondes are typically launched 4 times daily from SGP C1, nominally at 0530, 1130, 1730 and 2330

UTC each day (local time = UTC - 0600 hours). The PBL VAP uses three different algorithms for estimating $z_i$. These include the Heffter 1980, two bulk Richardson thresholds methods, and Liu and Liang (2010). The Heffter (1980) method is a well-established and widely used algorithm (e.g., Marsik et al. 1995, Delle Monache et al. 2004) that determines $z_i$ from potential temperature gradients using criteria related to the strength of an inversion and the potential temperature difference across the inversion. The bulk Richardson number ($Ri_b$) is a dimensionless number relating vertical stability to vertical shear. It represents

the ratio of thermally produced turbulence to that generated by vertical shear. Methods using $Ri_b$ to estimate $z_i$ assume that there is no turbulence production at the top of the stable boundary layer and therefore $Ri_b$ exceeds its critical value at the top of the boundary layer (Seibert et al. 2000). Several different critical thresholds of $Ri_b$, are provided in the literature based on resolution of sondes, location, etc. The ARM PBL VAP includes $z_i$ estimates based on two critical thresholds (0.25 and 0.5). Liu and Liang 2010 provide different thresholds for estimating convective and stable boundary layer depths using potential

temperature profiles. The inversion strength thresholds used in the method varies for given stability regime and land type classification (land, ocean, or ice). We note that the various estimates can differ considerably. More details about the ARM PBL VAP are provided by Sivaraman et al. (2013).



## 2.1 Preliminary Data Analysis

The Tucker method is used to estimate the height of the convective boundary layer from 30-minute vertical velocity variance profiles in the sgpdlprofwstats4newsC1.c1 VAP. For this study, we used a variance threshold of 0.04 m²s⁻², as determined in Tucker et al., 2009. The results are somewhat sensitive to the choice of threshold such that the $z_i$ estimates decrease as the threshold is increased (Berg et al., 2017).

     Figure 2 shows a typical example of vertical velocity variance during the warm season at SGP. Also shown are
estimates of $z_i$ from the Tucker method. We find that the Tucker method generally works well at tracking the height of the mixed layer during its initial development phase. In that case, there is sufficient SNR for the lidar to see both the developing convection and the overlying residual layer. However, as the mixed layer continues to deepen, at some point the SNR becomes too small to enable reliable estimates of the vertical velocity variance. This problem is sometimes compounded by a slight reduction in sensitivity during the hottest portion of the day which we suspect is the result of strong refractive turbulence in
the surface layer. Although this effect has not been thoroughly analyzed due to lack of refractive turbulence profiles and concurrent radiosonde data, there could be instances where the Doppler lidar is indeed measuring the top of the boundary layer. In any event, the loss of signal near the CBL top could result in $z_i$ estimates from the Tucker method to be low biased.

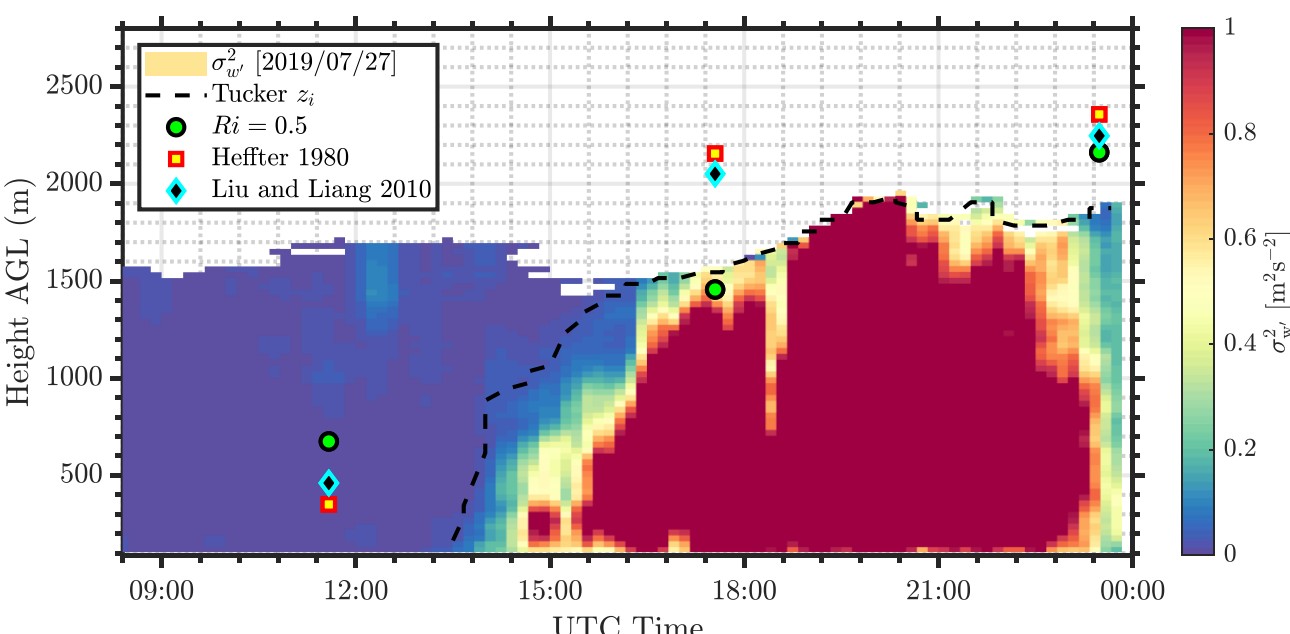

**Figure 2: Typical CBL showing lidar variance (colors), Tucker $z_i$ (black line), and radiosonde $z_i$ (three methods; symbols). The horizontal axis shows the number of hours after 00:00 UTC on 27 July 2019.**

     For this study, radiosonde-derived $z_i$ are assumed to be the best guess $z_i$ estimate and is used to calibrate the RF algorithm. Figure 3 shows comparisons between lidar-derived CBL (using the Tucker method) and simultaneous estimates





from the ARM PBL VAP. These comparisons were performed using 1785 days with daytime clear and shallow cumulus
conditions (identified as cloud base height less than 5 km from Doppler lidar and cloud fraction less than 0.1) for the years
2016 through 2019. From these results we found that the Liu and Liang (2010) technique resulted in the best overall agreement
with the lidar-derived estimates, in terms of the correlation coefficients (r = 0.75) and slope (0.70). Thus, $z_i$ estimates from the
Liu-Liang technique in the ARM PBL VAP (pblhtsonde1mcfarl.c) are used in this study as reference. It should be noted that
170    any of the above three model outputs can be used to calibrate the ML model, if needed. We also estimate night-time $z_i$
(assuming Liu and Liang 2010 estimates from radiosondes) at the SGP site using the RF model, but only to demonstrate the
capability of such a technique to reproduce the estimates as observed from radiosondes. In this paper, the primary focus is on
evaluating the daytime $z_i$ estimates from RF models.

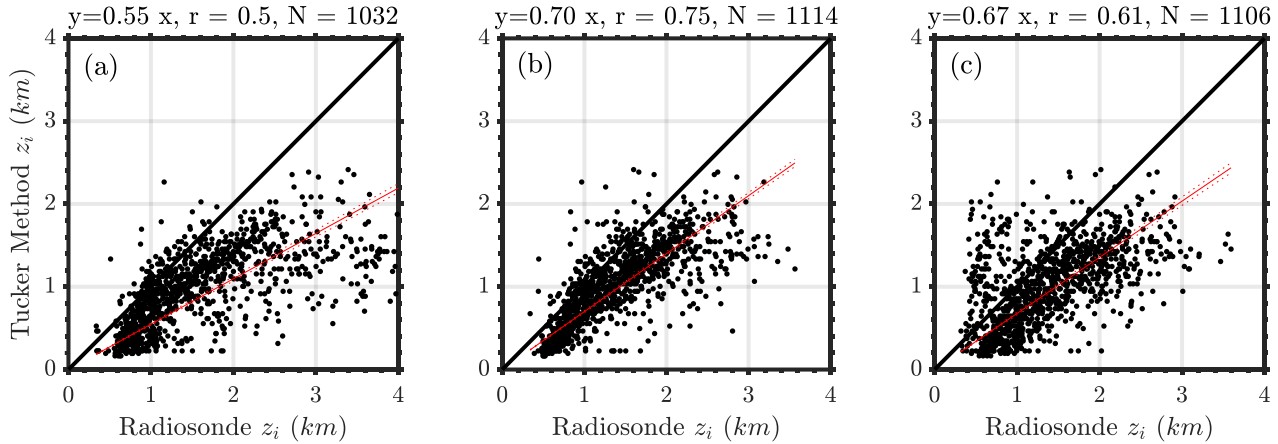

175

**Figure 3: Comparisons between Tucker method and three different $z_i$ estimates from radiosondes a) Heffter 1986, b) Liu and Liang
2010 and c) Bulk Richardson number method using a threshold of 0.5. The red line is the best fit with the fit y = mx shown above. r
is the correlation coefficient and N is the total number of radiosonde & Tucker method observations used in each scatter plot.**

**3 Machine Learning**

In this study, a RF algorithm was used to predict PBL heights. RF regression (Breiman, 2001) is an ensemble method
that is made up of a population of decision or decorrelated trees. Figure 4 provides a graphical illustration of the
RF bootstrapping process. Bootstrap aggregation (bagging) is used so that each tree can randomly sample from the dataset
with replacement, while only a random subset of the total feature set is given to each individual tree. The premise behind RF
is to improve the variance reduction of bagging by reducing the correlation between the trees, without increasing the variance.
The trees can be truncated to add further diversification. After construction, the population's individual predictions are
averaged to give a final prediction of the target variable. Ideally, this process results in a diversified and decorrelated set of
trees whose predictive errors cancel out, producing a more robust final prediction.





An advantage of RFs is their ability to determine the importance of all input features for the predictive process. This
is done by calculating the mean decrease in impurity, or the decrease in variance that is achieved during a given split in each
decision tree. The decrease in impurity for each input feature can be averaged over the entire forest, providing an approximation
of the feature's importance for the prediction (feature importance estimates sum to 100% to ease interpretability).  A sample
regression tree developed by real data used in this article is shown in Appendix A (Figure A1).

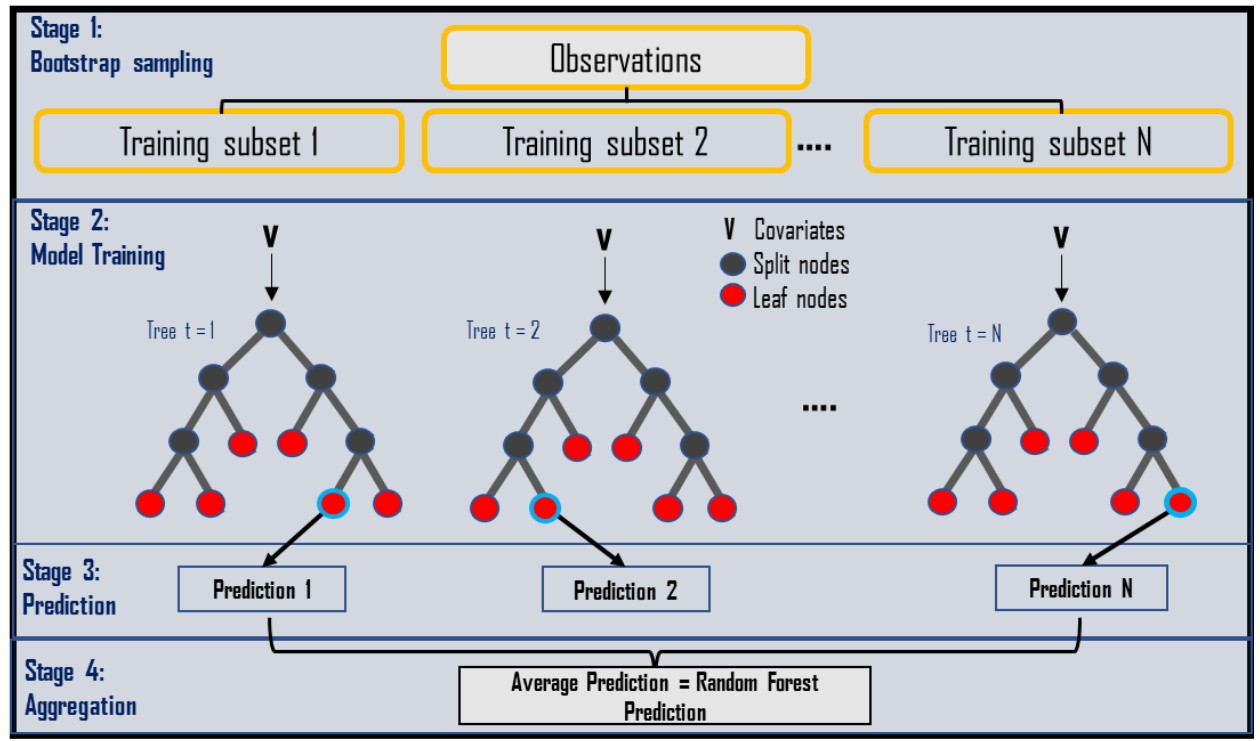


**Figure 4: Graphical representation of a standard RF algorithm (adapted from online sources and Figure A1).**

### 3.1  Model Hyperparameters

In this study a least-squares boosting regression (LSboost) ensemble RF model (Breiman 2001) is built based on
observational data listed in Table 1.  At every iteration, the ensemble fits a new decision tree to minimize the mean-squared
error between the observed response and the aggregated prediction of all decision trees developed previously.
Hyperparameters are used to control the learning process in the RF model.  It is good practice to tune hyperparameters such as
the maximum number of decision splits per tree, learn rate for shrinkage and the number of iterations for reducing the
generalization error (Breiman 2001).  In this article, hyperparameters for the RF model are chosen by performing a Bayesian





optimization on the data which minimizes the k-fold cross validation loss function for select hyperparameters. For this study, three hyperparameters were optimized: the number of tree splits, number of learning cycles or iterations, and learn rate for the model. Based on the optimization results, the number of iterations were set to 460, number of tree splits to 11 and the learning rate was set to 0.25 are used in the current model.

Regularization techniques are used to prevent statistical overfitting in a predictive model (Hastie et al., 2008), by reducing the magnitude of the coefficients of the RF model for certain parameters which do not contribute to the target variable. In general, regularization algorithms can treat issues such as multicollinearity and redundant predictors and make the model more precise. Lasso regularization is used in this study to avoid data overfitting (Tibshirani 1996).

**3.2  Data Preprocessing**

Surface and lidar data from 2016 to 2019 were used in this analysis. ARM VAPs provides processed and quality-controlled data from several atmospheric sensors. Each VAP provides quality control thresholds which are used to filter the data. For example, the lidar VAPs (DLWSTATS and DLWIND) provide quality control flags based on the system noise and signal to noise ratio (Newsom and Krishnamurthy 2020). For this analysis, thresholds specified in the VAPs are used to
remove any erroneous data. Similarly, quality control flags for all variables mentioned in Table 1 above are used to filter bad data (Tang et al., 2019). Since the temporal resolution of the surface data is variable, measurements are interpolated or averaged to the lidar 15-minute resolution time stamps. The frequency of radiosondes are generally 6 hours at SGP but can be in intervals of 3 hours during select field campaigns (Mather et al., 2018). For training purposes, surface and lidar data to the nearest radiosonde observations within 15-minutes are chosen.

Normalizing/standardizing/scaling processes are used to scale the variables in an ML model, such that they have the same order of magnitude in their variability. While in RF, whether data is scaled and normalized or not will generally make no significant effect on the model performance, but due to wide variations between site measurements (daytime and nighttime), the models showed slight improvements with standardization while estimating nocturnal $z_i$ estimates. Therefore, to improve nocturnal $z_i$ estimates, standardized datasets will be used in the following analysis for all conditions.

The RF model is trained using sub-surface, surface, lidar and concurrent radiosonde boundary layer height data from 2016, 2017 and 2018. Hereon this data is referred to as training-input features. A total of 3919 radiosonde $z_i$ measurements were used in the training process. The 2019 sub-surface, surface and lidar data are referred to as future-input features and are used to provide an independent dataset to evaluate the trained RF method. From an operational perspective, it is quite common to have missing input features. The most common approach is to fill-in (impute) the missing features (Hastie et al., 2008).
Surrogate data using a local median of the nearest 10 data points is used in the analysis and if more than 75% of data are not available a *NaN* is entered (Hastie et al., 2008). It is critical for the RF model to deal with missing values in its training phase. In some cases, the added noise in the system can help improve the stability of the RF model. But too much noise is also detrimental in the training process, and the imputed data may no longer be useful and can cause erroneous results. The

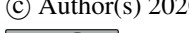



imputation process is done inherently during the RF bagging process. The effect of data imputation on model performance
was evaluated by training the RF model with and without data imputation. Four different possibilities in evaluating the
performance of the RF model with and without imputation are evaluated, as shown in Table 2 below. In the case of no
imputation (no missing data), only time periods when all features/parameters are available are used in the RF regression. In
the case of 50% missing data, approximately 50% of the time series data were missing (at random) due to say issues with data
quality and surrogate data was used. The choice of 50% was to mimic a worst-case real-world scenario based on experience,
where either one or two instruments from the feature set of 20 odd variables have data quality issues. Other combinations,
such as no missing data used in the RF regression during training process and 50% future-input feature missing data and vice
versa were evaluated.

**Table 2. Data imputation evaluation and performance on boundary layer heights for year 2019**

| Case | Training-Input Features | Future-Input Features | $R^2$ (%) | RMSE (m) | $Y_{RF} = m\ X_{RS} + C$ |
|---|---|---|---|---|---|
| No Imputation | No missing data | No missing data | 76.86% | 324 | $Y_{RF} = 0.912\ X_{RS} + 211$ |
| Future-Input Imputation | No missing data | 50% missing data imputed | 65.28% | 366 | $Y_{RF} = 0.789\ X_{RS} + 288$ |
| Training-Input Imputation | 50% missing data imputed | No missing data | 76.15% | 328 | $Y_{RF} = 0.876\ X_{RS} + 258$ |
| Training & Future-Input Imputation | 50% missing data imputed | 50% missing data imputed | 66.50% | 357 | $Y_{RF} = 0.792\ X_{RS} + 324$ |

* RF – Random forest, RS – Radiosondes

As it can be seen from Table 2, data imputation overall reduces the accuracy of RF model performance. Missing future-
input features seems to have the highest effect on the RF model $z_i$ estimates, regardless of training the model based on missing
training-input features. This could be due to several reasons, such as the median value does not represent the current state of
the missing data, the same input features are not missing in both training and future input features as the combinations to test
are near infinite (as the data in real-world can be missing in random as in the case above), etc. Therefore, in this analysis, the
model is trained with no missing data, and no imputation is done on the data (either input or future features) to accurately test
the efficacy of the RF model. Future studies are planned to implement a better imputation model based on data from past
trends for a given feature and test RF model performance.
RF models are generally site specific, initial tests (not shown) show the possibility to develop a generalized RF model to
estimate $z_i$ for all sites around SGP (C1 and other satellite sites) with good accuracy under all atmospheric conditions. A
round-robin type analysis, where in a model developed at a given site is tested at every other site and vice versa would be a
valuable exercise (Bodini and Optis 2020).





**4 Performance of RF $z_i$**

Boundary layer height predictions from the RF model and Tucker method were compared to radiosonde estimates. Mean absolute error (MAE) is defined as:

$$MAE = \frac{1}{N}\sum_{i=1}^{N}\left|z_{iRS} - z_{i\gamma}\right| \tag{1}$$


Where, $z_{iRS}$ is the boundary layer height estimated from the radiosondes (Liu and Liang 2010), $z_{i\gamma}$ is the boundary layer height estimated from either the Tucker method or RF model. The root mean square error is defined as

$$RMSE = \sqrt{\frac{1}{N}\sum_{i=1}^{N}\left(z_{iRS} - z_{i\gamma}\right)^2} \tag{2}$$


Similarly, the linear correlation coefficient ($R$) is defined as

$$R = \frac{1}{\sigma_{RS}\sigma_{\gamma}(N-1)}\sum_{i=1}^{N}(z_{iRS} - z_{RS})(z_{i\gamma} - z_{\gamma}) \tag{3}$$

Where, $z_{RS}$ and $z_{\gamma}$ denote the mean of radiosonde and RF/Tucker method boundary layer heights, respectively, $\sigma_{RS}$ and $\sigma_{\gamma}$ denote their standard deviations, and $N$ denotes the number of samples.

MAE and RMSE for the daytime atmospheric conditions from Tucker and RF methods is shown in Table 3. Daytime is defined as period when surface heat flux is positive (i.e., sunrise to sunset), clear sky is defined as time periods when no clouds are observed from the Doppler lidar (with a cloud fraction less than 0.1) and cloudy conditions are defined as time

periods when clouds below 5 km are observed from the Doppler lidar (with a cloud fraction greater than 0.1). It can be observed that the RF method shows considerable improvements compared to the Tucker method for all the three categories. Improvements of 40-50% in MAE and RMSE are observed under various conditions. The least improvement in RF $z_i$ estimates is observed during cloudy conditions, with a MAE improvement of 45% compared to Tucker method. Correlation coefficients are also observed to improve significantly during all daytime conditions.

Due to the presence of a nocturnal low-level jet (LLJ) at SGP, all seasons, the radiosonde night-time $z_i$ estimates are generally below the LLJ height and is well tracked by the RF model (see Section 4.1). Since the Tucker method is not applicable during night-time conditions, a comparison of the RF model improvement with respect to other remote sensing devices that continuously monitor the boundary layer height needs to be conducted (e.g., Raman lidar) and is a part of future work.






**Table 3. Systematic mean absolute errors, root mean square error and correlation coefficient ($R^2$) between RF, Tucker method and radiosonde $z_i$ estimates in 2019.**

| Observed atmospheric conditions | MAE (m) | | | RMSE (m) | | | $R^2$ | |
|---|---|---|---|---|---|---|---|---|
| | RF | Tucker Method | % Improvement | RF | Tucker Method | % Improvement | RF | Tucker Method |
| Daytime Only | 167 | 311 | 46% | 249 | 441 | 43% | 0.845 | 0.545 |
| Daytime Clear sky | 165 | 336 | 51% | 235 | 479 | 51% | 0.857 | 0.520 |
| Daytime Cloudy | 141 | 255 | 45% | 208 | 363 | 43% | 0.878 | 0.725 |

Nocturnal estimates of $z_i$ from radiosondes at the SGP are much more uncertain (Sivaraman et al., 2013). Regardless

of the accuracy of the $z_i$ estimate from radiosondes, the ability of the RF model to predict the input data is evaluated. Consequently, the data was split into five atmospheric conditions, 1) daytime and clear sky (26%), 2) day and night clear sky (60%), 3) day and night cloudy (40%), 4) daytime only (53%) and 5) night-time only (46%). Although some cases overlap in the above situations, the idea was to evaluate the RF model performance reasonably during both day, night-time, cloudy and clear-sky condition combinations. These five atmospheric conditions were chosen, since these are commonly observed

atmospheric conditions at SGP and have observed to have an impact on $z_i$ (e.g., variations in cloud base height varies $z_i$). The associated correlation plots between RF $z_i$ estimates and corresponding radiosonde $z_i$ estimates is shown in Figure 5. Overall, the RF model $z_i$ estimates correlate well with radiosonde $z_i$, with an $R^2$ of greater than 0.85 during all the above conditions. During clear sky conditions (including both daytime and night-time), RF $z_i$ estimates shows the highest correlation of 0.88. Although the sample size varied for five categories, the correlation coefficients are overall high for the RF $z_i$ estimates

(compared to Tucker method in Figure 3). During the night, the $z_i$ values from the radiosondes are relatively constant, and the RF estimates are consistent with these values. However, due to the small dynamic range of the night-time $z_i$ values, the correlation between the radiosonde and RF methods is relatively poor (0.54). It is important to note that the authors are only demonstrating the ability of the RF model to replicate night-time $z_i$ estimates compared to radiosonde-derived values and are not making any judgements on the accuracy of "true" night-time $z_i$ at SGP, which could be different than the radiosondes

estimates that used to calibrate the model. Based on the slopes of the regression lines in Figure 5, there seems to be a tendency of the RF model to overestimate $z_i$ when it is small and slightly underestimate $z_i$ when it is large. Possible reasons for this trend are under further investigation.

Overall, a uniform improvement in $z_i$ can be obtained using RF techniques. Reducing or increasing the training data had an impact on the RF model performance, but the increase in the magnitude of the correlation coefficient was negligible

after 2 annual cycles of data. In this analysis, three years of data was used for training the RF model.





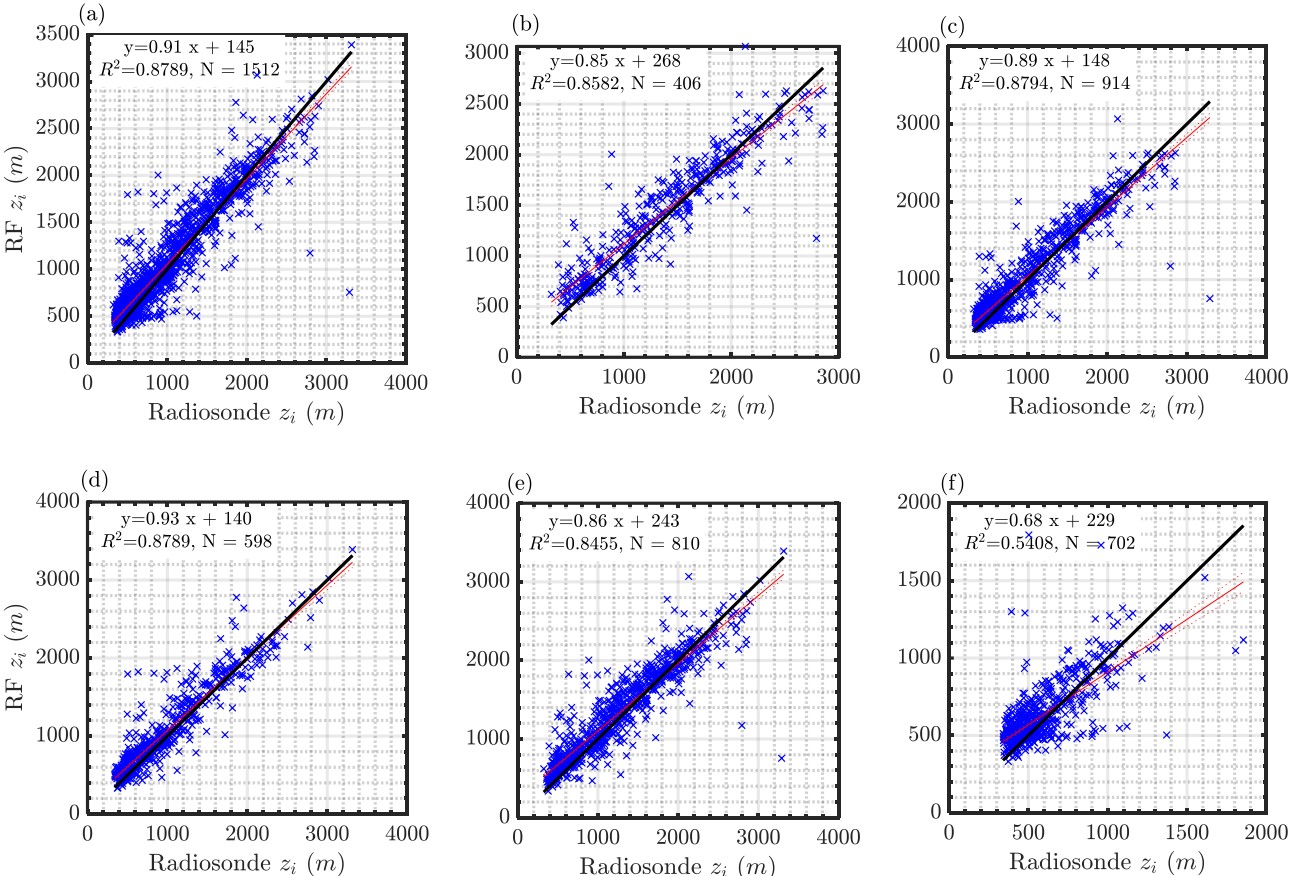

**Figure 5: Correlations between RF PBL height and Radiosonde PBL height for a) all data in 2019, b) daytime clear sky, c) clear sky, d) cloudy, e) daytime only and f) night-time only.**


## 4.1 Time series, diurnal and seasonal performance

The RF model was trained using data from 2016 to 2018, and $z_i$ estimates for 2019 were estimated using the parameters listed in Table 1. Figure 6 and Figure 7 show lidar $z_i$ estimates from Tucker method, radiosondes and RF model on June 20 and June

22, 2019, respectively. Cloud base height and vertical velocity variance profiles from Doppler lidar are also overlaid. Due to an ongoing field campaign, the micropulse differential absorption lidar (MPD) demonstration project, these days observed a higher frequency of radiosonde observations (Weckwerth et al., 2020). It is clear the RF model $z_i$ closely follows radiosonde $z_i$ estimates and the Tucker method underestimates $z_i$ as estimated from radiosondes. Although, the Tucker method is observed to track the convective boundary layer height effectively, a bias is observed when compared to the radiosonde $z_i$. Optimizing the vertical velocity variance thresholds could potentially reduce the bias in certain conditions, but the bias is not uniform

across all time periods (see Figure 7). Since aerosol concentration decreases with altitude, signal availability reduces as a





function of height. During peak convection, when the aerosol concentration above the boundary layer is minimal, lidar measurements sometimes do not reach the top of the boundary layer with sufficient signal-to-noise ratio to be detected. The lidar beam also attenuates considerably when it encounters clouds or fog, due to increased atmospheric scattering or attenuation. In Figure 6, it can also be seen that lack of aerosols limits the Doppler lidars' ability to measure above the

boundary layer height during peak convective periods. During night-time conditions, vertical velocity variance is low and is not effective in estimating $z_i$. In this study, the lowest range-gate is used as the stable boundary layer height from the lidar as a first guess. As mentioned earlier, nocturnal conditions during Summer months are dominated by southerly winds where in the nocturnal boundary layer is capped by a low-level jet (Krishnamurthy et al., 2020). During these conditions, the RF model shows near constant $z_i$ which is observed to be well correlated with radiosonde $z_i$ (just below the LLJ height).


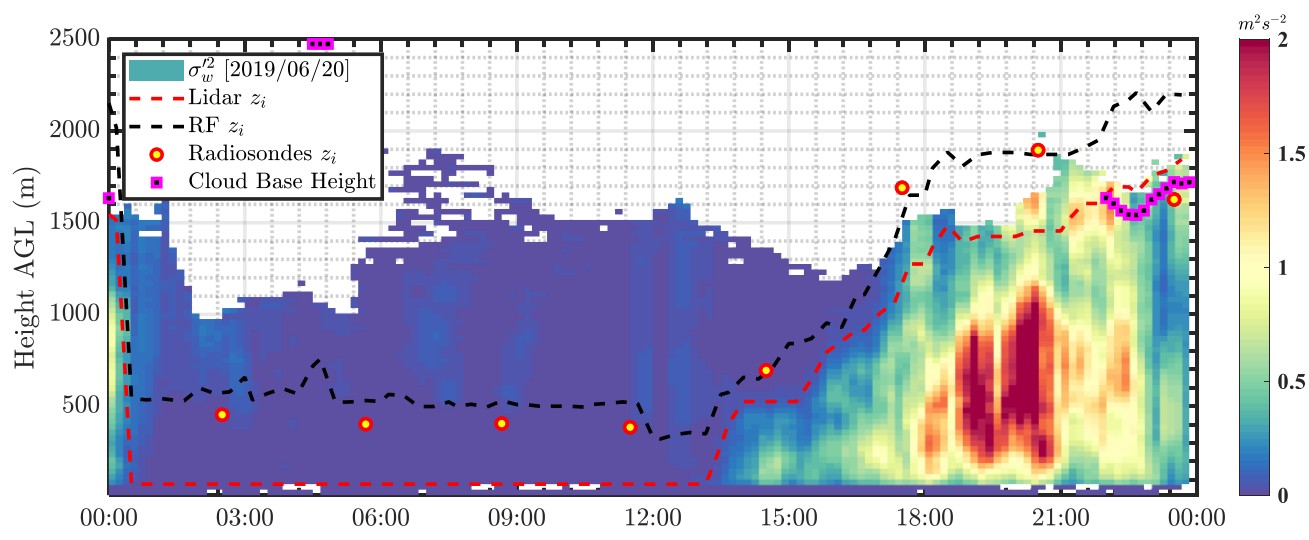

**Figure 6: Boundary layer height estimates at SGP central facility on June 20, 2019 from lidar thresholding technique (Tucker et al., 2009 for convective $z_i$), RF model $z_i$, radiosondes $z_i$ (Sivaraman et al., 2013), cloud base height estimates from lidar (Newsom et al., 2019) and the background colors represent vertical velocity variance measurements from Doppler lidar.**


In Figure 7, the effectiveness of the RF model can clearly be observed. At 0300 hours & 0600 hours UTC, during stable nocturnal conditions, the RF model matches the radiosonde estimates very well. At 0900 hours UTC, a possible nocturnal Convection Initiation (CI) event results in high vertical velocity variance for several hours (Reif et al., 2017). CI refers to the process in which an air parcel is successfully lifted to its level of free convection and produces a precipitating

updraft (Markowski and Richardson 2010). The RF model is observed to detect that burst of convection and provide coherent boundary layer heights past 1200 UTC until daytime transition at ~1400 UTC. The correlation between RF model and radiosonde $z_i$ estimates is very high. Therefore, various atmospheric interaction effects are aptly characterized by the parameters in the RF model. Hourly averaged $z_i$ and daily maximum $z_i$ averaged for each season in 2019 from Tucker method,




RF model and radiosondes are shown in Figure 8. Although, the number of samples between the radiosondes and RF model

estimates are vastly different, the generic trend in the hourly and seasonal boundary layer height variability is well captured

by the RF model. Although the Tucker method captures the average boundary layer height trend, it shows a clear bias in

convective boundary layer height estimates compared to radiosonde and RF model derived $z_i$ values. As observed earlier from

the time series analysis, a standard bias correction would not always improve $z_i$ estimates from the Tucker method. Daily

daytime maximum $z_i$ estimates averaged over 4 seasons in 2019 from all three methods is shown in Figure 8b. Summer months

(May through August) show high boundary layer heights. During these months, the peak convective period occurs during the

daytime at around 2000 hours UTC, and the average boundary layer height as observed from the RF model is ~2100 m above

ground level which correlates well with radiosondes released during the same time period. During winter and fall months, the

peak convective period does not always occur at 2000 hours UTC and therefore the maximum $z_i$ estimates from radiosondes

do not coincide with RF estimates. The Tucker method invariably underestimates maximum $z_i$ during all seasons.


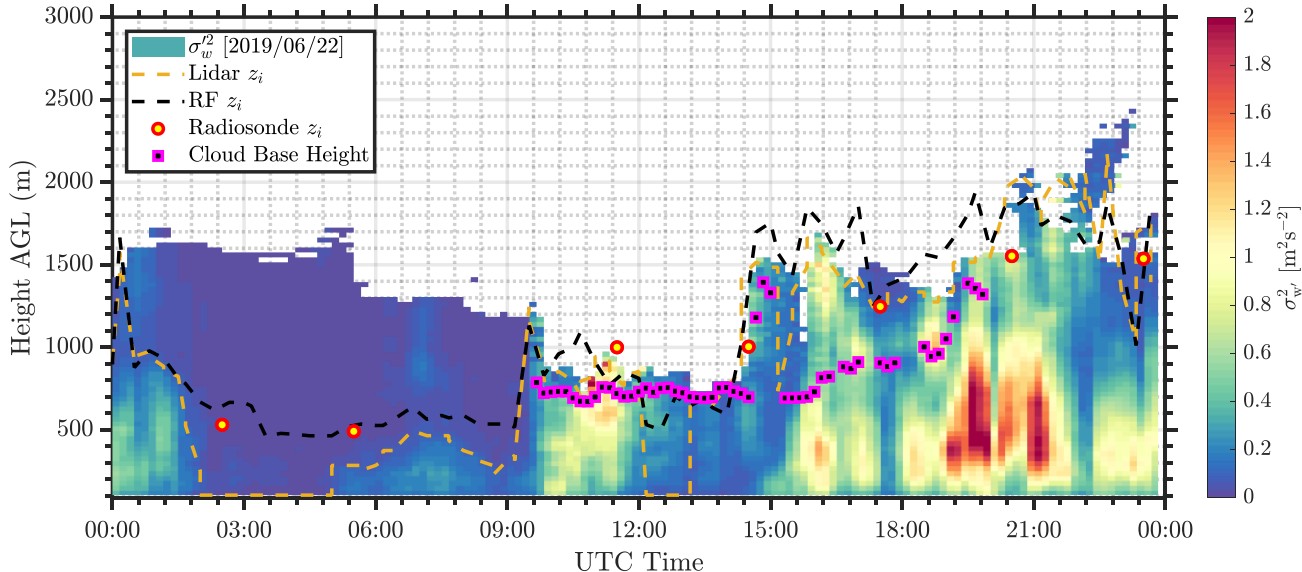

**Figure 7: Boundary layer height estimates at SGP central facility on June 22, 2019 from lidar thresholding technique (Tucker et al., 2009 for convective zi), RF model zi, radiosondes zi (Sivaraman et al., 2013), cloud base height estimates from lidar (Newsom et al., 375 2019) and the background colors represent vertical velocity variance measurements from Doppler lidar**



(a)



**Figure 8: a) Hourly averaged $z_i$ estimates at SGP central facility for 2019 from RF, Tucker method, and radiosondes. Total number of samples (N) for each dataset is also shown in the legend. b) Daily daytime maximum $z_i$ estimates for 4 seasons (DJF, MAM, JJA, SON) from RF, radiosonde, and Tucker method.**

## 4.2 Input Feature Importance

All the input features within the RF model explain approximately 82% of the total variance in the data. Table 4 provides the unbiased predictor importance estimates, which is computed by permuting or shuffling a variable in the model and estimating its mean square error (Breiman 2001). If a predictor is significant in prediction, then permuting its values will affect the model error and vice versa. However, if two input variables are highly correlated (as is expected when testing





atmospheric forcing), it is highly unlikely that the reported importance values will accurately represent each variable's significance (Breiman, 2001). Based on this analysis, the most important parameter is the initial $z_i$ guess from the Tucker method. This provides a very good first guess to the RF model, especially during convective conditions. Although, the RF

model is sensitive to the initial guess from the Tucker method, it is observed to be robust enough to ignore uneven spikes in $z_i$ estimates due to noise in the lidar vertical velocity variance data (Figure 7). The second most important feature observed to have high correlation with $z_i$ estimate is the hour of the day. A clear diurnal pattern in $z_i$, i.e., higher values in the daytime and near constant values during the night-time, estimates are observed at SGP. Therefore, hour of the day can effectively classify the data, which is beneficial in the RF bagging process. Relative humidity also shows higher importance (> 5%), where drier

conditions are observed to have higher correlations with boundary layer height. Deep convection and larger $z_i$ is generally a consequence of greater sensible heat flux and lower latent heat flux, which are primarily due to higher surface temperature and lower relative humidity. Based on an evaluation of $z_i$ over Europe, it was also observed that relative humidity had a strong negative correlation with $z_i$, and surface temperature had a positive correlation (Zhang et al., 2013). Other features such as lidar attenuated backscatter, surface air temperature, Monin-Obukhov length, soil temperature, surface wind direction, and

TKE all are observed to be important for accurately characterizing the boundary layer height at SGP. Other surface features such as surface friction velocity, sensible heat flux, longwave radiation, etc., have lower correlations with $z_i$ within the RF model framework. Therefore, the model can be reduced to the list of parameters defined in Table 4 for optimal estimation of the boundary layer height at SGP.

**Table 4: Key parameter/feature unbiased importance estimates during all conditions**

| Parameters/Features | % Importance |
|---|---|
| Tucker method $z_i$ | 58.67% |
| Hour of the day | 10.05% |
| Surface Relative Humidity | 6.82% |
| Attenuated Backscatter | 2.90% |
| Surface Air Temperature | 2.77% |
| Monin-Obukhov Length | 2.77% |
| Soil Temperature | 1.92% |
| Surface Wind Direction | 1.78% |
| Turbulence Kinetic Energy | 1.32% |
| Others | < 11% |

To assess the key features during night-time, a similar RF model was built by conditionally sampling night-time data. During night-time, the key parameters that impact the RF model predictions are shown in Table 5. It is interesting to note that the key features deemed important during night-time are significantly different compared to all conditions, and the percent

importance is more evenly distributed across many features. This alludes to the fact that nocturnal stable boundary layers are indeed complex to model and several processes are at play (Fernando and Weil 2010). Monin-Obukhov length is observed to





have the highest impact on the night-time RF model estimates and is consistent with theory on stable boundary layers (Zilitinkevich 1972, Zilitinkevich and Baklanov 2002). Although night-time $z_i$ initial guesses are generally a constant (lowest lidar range-gate if no high vertical velocity variance is observed), the initial guess has shown to be effective in adjusting the

RF model $z_i$ estimates. Other local parameters such as soil temperature, surface air temperature, dew point temperature, longwave radiation and turbulence kinetic energy are observed to be correlated with night-time $z_i$ estimates. One of the stable boundary layer models by Brost and Wyngaard 1978 is given by:

$$z_i = 0.4 \left( \frac{u_* L}{|f|} \right)^{1/2} \qquad (1)$$


Where, $u_*$ is the friction velocity, L is the Monin-Obukhov length and f is the Coriolis parameter. As shown in Table 5, the night-time parameters deemed important by the RF model include both Monin-Obukhov length and friction velocity. As discussed earlier, the dominant feature of nocturnal boundary layer at SGP is the presence of the LLJ. The turbulence production at SGP is not only influenced by surface characteristics but also heavily influenced by the presence of the LLJ. A

preliminary comparison with the above model to RF model $z_i$ estimates at SGP was very poor, as the radiosondes (from all three methods) invariably pick up the nocturnal low-level jet at SGP as the height of the boundary layer. Although the height of the nocturnal boundary layer height could be debatable at SGP, the premise of this paper is to show the effectiveness of the RF model in tracking and detecting the boundary layer height as well as the input boundary layer height provided to the model. Further research needs to be conducted on providing widely acceptable nocturnal boundary layer height at SGP, and a trained

RF model can provide continuous boundary layer height estimates even in nocturnal conditions with acceptable levels of accuracy.

**Table 5: Key parameter/feature unbiased importance estimates during night-time conditions**

| Parameters/Features | % Importance |
| --- | --- |
| Monin-Obukhov Length | 19.12% |
| Lidar-only $z_i$ | 11.49% |
| Soil Temperature | 7.81% |
| Surface Air Temperature | 7.56% |
| Dew Point Temperature | 6.31% |
| Longwave radiation | 6.24% |
| Turbulence Kinetic Energy | 5.21% |
| Net radiation | 4.72% |
| Surface wind speed | 4.28% |
| Surface wind direction | 3.76% |
| Lidar Dissipation rate variance | 3.54% |
| Surface friction velocity | 3.45% |
| Cloud base height | 3.40% |
| Shortwave radiation | 3.26% |





| Others | < 10% |
|---|---|

The partial dependence of key features on $z_i$ during all conditions is shown in Figure 9. Partial dependence estimates show the

marginal effect of features on the predicted outcome of a ML model. Therefore, a higher partial dependence estimate corresponds to higher sensitivity to the predicted outcome, in our case the boundary layer height and vice versa. From Figure 9, we see that RF model $z_i$ is sensitive to warmer soil temperatures, lower relative humidity conditions, daytime hours, higher $z_i$ from Tucker method, northerly wind directions, and stable atmospheric conditions. Most of these conditions would mimic dry convective conditions, with increased turbulence activity within the boundary layer. Monin-Obukhov length is observed

to effectively categorize the training data into stable and unstable atmospheric conditions, with high partial dependence estimates during stable boundary layer conditions. Similar relationships can be derived for other parameters. It is important to note, that the parameters shown to be important are with respect to the RF model, are features that successfully aide in the RF bagging process. Santenello et al., 2007 and Tang et al., 2018 showed parameters such as soil moisture and evaporative flux to be key variables in warm seasons, in a 2-parameteric regression framework, to affect boundary layer properties. Within

the RF model, although those parameters are deemed important, soil temperature, lidar backscatter and relative humidity were shown to have a higher impact on boundary layer height at each site.

In this research, we have mostly analysed using standard processed data from SGP instruments as an input into the RF model. A smarter way to approach would be by providing several non-dimensional inputs (such as Bulk Richardson number, Froud Number, etc) as inputs which would capture multiple dimensions of the data with a single variable (Vassallo

et al., 2020). Since non-dimensional scaling is a common approach in atmospheric fluid dynamics to detect patterns in the data, a similar approach would provide the RF model with various relations and be helpful in classifying the data better. But further research needs to be conducted in defining the key non-dimensional parameters that affect $z_i$, and is a part of future work.





**Figure 9:** RF partial dependence during all conditions from a) Tucker method $z_i$, b) relative humidity c) hour of the day, d) Monin-Obukhov Length, e) surface wind direction and f) soil temperature to boundary layer height at the central facility. High dependence shows more sensitivity of the RF model to the bin of feature values.

## 5 Normalized vertical velocity variance profiles

Within a convective boundary layer, vertical velocity variance profiles are often scaled by the convective velocity scale (w*), which is a function of $z_i$ (Lenschow et al., 1980) for analysis. Therefore, any error in boundary layer height estimates can result in altering the vertical velocity variance profiles. Herein, we attempt to estimate the effect of boundary layer height on normalized vertical velocity variance profiles, often used in atmospheric models. The convective velocity scale is given as



$$w^* = \left[\frac{g z_i \overline{w'\theta'}}{\theta}\right]^{1/3} \tag{2}$$

where, g is the gravitational constant, $\theta$ is potential temperature, and $\overline{w'\theta'}$ is heat flux. The heat flux is obtained from the surface Eddy Covariance (ECOR) system at SGP C1 (Cook 2016) and potential temperature is measured at 4-m. An

uncertainty in $z_i$ will cause a non-linear effect in the convective velocity scale estimates. Assuming the uncertainty in $z_i$ can be given as $z_i'$, and the mean is given as $\bar{z}_i$. The error caused in the convective velocity scale due to uncertainty in $z_i$ can be formulated as shown below:

$$w^* = \left[\frac{g(\bar{z}_i + z_i')\overline{w'\theta'}}{\theta}\right]^{1/3} \tag{3}$$

This can also be written as

$$w^* = \left[\frac{g\bar{z}_i \overline{w'\theta'}}{\theta}\right]^{1/3} \left(1 + \frac{z_i'}{\bar{z}_i}\right)^{1/3} \tag{4}$$

Therefore, the convective velocity scale error due to uncertainty in $z_i$ can be estimated using the term $(1 + x)^{1/3}$, where $x = $

$\frac{z_i'}{\bar{z}_i}$. Based on observations at SGP C1, $z_i$ from Tucker method is observed to be negatively biased to radiosonde estimates. The ratio, $(\frac{z_i'}{\bar{z}_1})$, is calculated using the median error between Tucker method and RF model $z_i$ by the median $z_i$ using the RF model. The ratio is calculated to be approximately -0.28, for data from 2016 to 2019. This would result in an uncertainty of approximately 10% in the convective velocity estimates when Tucker method $z_i$ values are used in the calculations. Although this is an average, during certain conditions (e.g., during transition time periods) the effect of poor characterization of $z_i$ can

be even larger. Figure 10 shows average vertical velocity variance profiles during convective time periods at SGP from 2015 to 2019 using $z_i$ estimates from Tucker method and RF model. Higher $z_i$ values result in lower scaled vertical velocity variance estimates. Differences in variance profiles are observed to be smaller during daytime transition (1500 hours UTC or 0900 hours local time), as the Tucker method generally provides reliable $z_i$ estimates. While during peak convective conditions (2000 hours UTC) and evening transition (2300 hours UTC), as observed earlier, the Tucker method $z_i$ estimates tend to diverge

from radiosonde $z_i$ depending on the scenario and are negatively biased. Overall, the error introduced in the scaled vertical velocity variance profiles due to Tucker method $z_i$ are above 15%.

As per definition, the turbulence at the top of the boundary layer is expected to be near zero. It can be observed in Figure 10, during morning transition periods, when the lidar does measure above the boundary layer, the scaled vertical velocity variance profiles do converge to zero. While during peak convective conditions the scaled profiles do not converge to zero

and are observed to have an offset of generally around 0.1 at $z_i$ (also observed in Lareau et al., 2018). This could be due to





higher uncertainty in lidar vertical velocity variance estimates near $z_i$, ineffective filtering of the lidar false alarm rates, uncertainty in surface in-situ measurements and the possibility of residual turbulence above the boundary layer during downdrafts/updrafts.

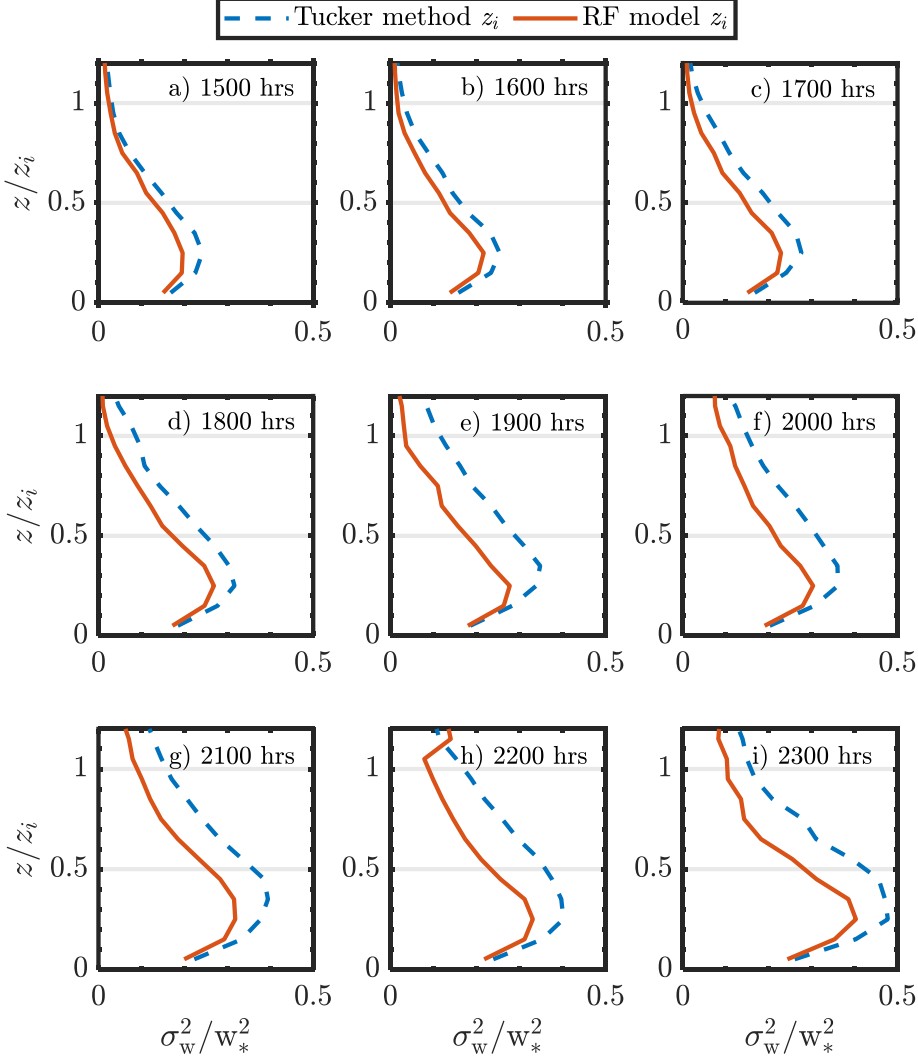


**Figure 10: Average normalized vertical velocity variance profiles during convection periods [1500 UTC to 2300 UTC (a-i)] at SGP from 2015 to 2019 versus non-dimensional height using Tucker method and RF model estimated $z_i$. Appropriate $z_i$ (RF model or Tucker method) was used in both X and Y axis scaling.**



## 6 Case Study: Preliminary Model Comparisons

The surface and sub-surface layer measurements are key for understanding land-atmosphere interactions. Land–atmosphere interactions drive Earth's surface water and energy budgets. They can alter clouds and precipitation around a region, affect the growth of the planetary boundary layer height, and can influence the persistence of extremes such as droughts. In view to better understand land-atmosphere interactions, a field campaign, Holistic Interactions of Shallow Clouds, Aerosols and Land Ecosystems (HI-SCALE), was conducted at SGP site in Oklahoma (Fast et al., 2019). The field campaign was conducted at SGP from April to September of 2016, with two 4-week intensive observational periods in May and September. Simulations were conducted on select clear sky days during the HI-SCALE filed campaign. Two simulations using different modelling systems were performed, the Energy Exascale Earth System Model (E3SM; Golaz et al., 2019) Atmosphere Model version 1 (EAMv1; Rasch et al. 2019), and a Large-Eddy Simulation (LES) model. The Weather Research Forecasting (WRF, Skamarock et al. 2008) LES is set up in the same way as that used in operational LASSO (Gustafson et al., 2017, 2020). The WRF model version used is v3.7. The model horizontal domain is square, doubly periodic and 25.6 km wide with a 100-m horizontal resolution. The model top is set at 14.8 km above the surface. There are 226 vertical levels with a vertical grid spacing of ~30 m in the lowest 5 km. The model is run for 15 hours for each case day starting from 6 AM. The Rapid Radiation Transfer Model for Global Climate Models (RRTMG) parameterization is used for shortwave and longwave radiation (Clough et al. 2005; Iacono et al. 2008; Mlawer et al. 1997). The Thompson parameterization is used for microphysics (Thompson et al. 2004; 2008). The Deardorff 1.5-order turbulent kinetic energy (TKE) approach is used for subgrid-scale parameterization (Deardorff 1980). The model is initialized with ARM sounding from the SGP site (ARM user facility, 2001). The surface sensible and latent heat fluxes are horizontally uniform and prescribed from the ARM constrained variational analysis (VARANAL) data product (ARM user facility, 2004). The large-scale forcing is also taken from the ARM VARANAL data product. Due to high computational expense, the LES model, was run for three days while EAMv1 model was run for the entire duration of the HISCALE campaign. The EAMv1 model is run in the standard coarse-resolution configuration with ~ 1-degree horizontal grid spacing and 72 vertical levels and a physics timestep of 30 minutes and a cloud and turbulence timestep of 5 minutes. In these models, $z_i$ was estimated using resolved vertical velocity variance estimates from the LES or the parameterized vertical velocity estimate from the Cloud Layers Unified By Binormals (CLUBB) boundary-layer parameterization applied in E3SM (Golaz et al., 2002; Larson and Golaz, 2005; Bogenshutz et al., 2013; Larson, 2017), and like the Tucker method a low threshold was used to estimate the depth of the convective boundary layer. Nocturnal $z_i$ is not estimated using model data.

Figure 11 shows Doppler lidar vertical velocity variance measurements for three days during the HISCALE campaign (September 10 – 12, 2016) and boundary layer height estimates from RF model, radiosondes, LASSO, and E3SM. Since RF model provides $z_i$ estimates at a much finer temporal resolution than radiosondes, these estimates are ideal for comparing with models and assessing areas where the model performance can be improved. Therefore, in this preliminary comparison, the primary motivation is to see if we can identify areas where the models diverge significantly by using near continuous accurate





$z_i$ values from the RF model. Due to computational expense, we could evaluate only three days of model results, but further research is needed in performing a thorough evaluation.

September 10th was relatively calm with northerly winds and no clouds were observed during daytime. September 11th & 12th observed southerly winds and clear sky conditions during both day and night-time. Daytime maximum surface air temperature is observed to increase progressively from September 10 to 12, and as mentioned earlier higher air temperature results in deeper planetary boundary layer due to increased convection (see Figure 11).

    Overall, the LES model compares better to RF estimates, while EAMv1 is observed to underestimate $z_i$. Due to the
coarse resolution of EAMv1, ~ 100 km, $z_i$ estimates are averaged over a large domain and do not generally capture the fine scale variability. The LES model although is observed to pick up morning transition, it is observed to diverge from observations during evening transitional periods and does not capture the decay of turbulence accurately. During peak convective conditions, when the vertical velocity variance is large, the LES is observed to correlate very well with RF model and radiosonde $z_i$. Although EAMv1 is observed to mostly underestimate $z_i$ compared to RF model estimates, occasionally
the model captures the peak convective trends. Similar to LES model $z_i$, the EAMv1 also is not observed to capture evening decay of turbulence accurately but is observed to not track the early morning transition at SGP as well. Such systematic differences between the model and data is crucial for targeting future research directions. Further study is needed to evaluate on reasons why models tend to deviate during early morning and/or evening transition periods.

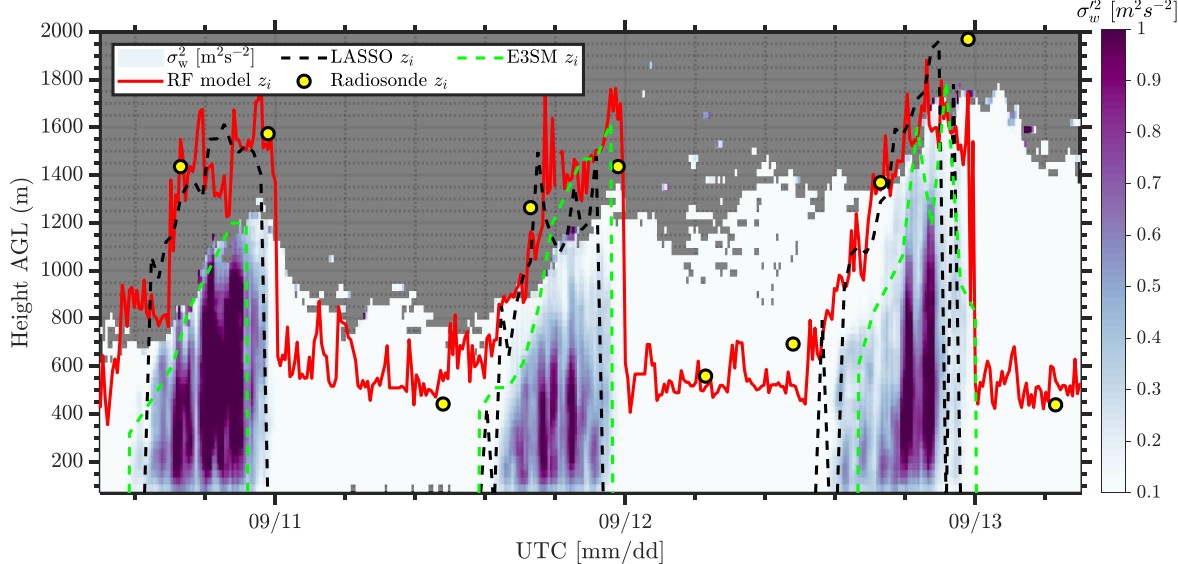


**Figure 11: Vertical velocity variance ($\sigma_w^2$) estimates from Doppler lidar for three days [September 10th to 12th, 2016] with $z_i$ estimates from a) RF model (red solid line), b) Radiosondes (yellow circles), c) LASSO model (black dashed line), and d) E3SM model (green dashed line).**





## 6 Summary

This study used a range of near surface, sub-surface and Doppler lidar parameters to predict boundary layer heights at ARM SGP site using a RF model. The RF model was trained using several years of data, and the model was validated with radiosonde estimates of boundary layer height. Since the Tucker method is observed to be low biased during peak convective periods due to low signal to noise ratio of the Doppler lidars as the boundary layer deepens, the RF model corrects for the bias.

Seasonal and diurnal variations of $z_i$ as observed from radiosondes correlate well with RF model $z_i$. During convective boundary layer conditions, the mean absolute error of boundary layer height estimated by RF model reduced by almost 50% compared to Tucker method. Significant improvement was also observed during clear sky, and cloudy conditions. Nocturnal estimates from RF model were not well correlated with radiosonde measurements, mostly due to near constant estimates of nocturnal boundary layer height at SGP (due to the presence of the LLJ). Moreover, valuable information on the impact of

surface parameters on nocturnal $z_i$ estimates by RF model provides avenues for further research in accurately estimating stable boundary layer heights at SGP. The key variables that have shown to have the largest impact on the RF model predictions are the initial guess of the boundary layer height from Tucker method, hour of the day, surface relative humidity, soil temperature, and attenuated backscatter (aerosol loading). During nocturnal conditions, several parameters, such as the Monin-Obukhov length, soil temperature, surface air temperature influence the RF model estimates. These parameters are aligned with

theoretical parameterization schemes used to estimate boundary layer heights. The RF model used in this study explains around 82% of the variance in the data at SGP C1.

Uncertainty in convective boundary layer heights results in more than 10% difference in convective velocity scale estimates when used Tucker method. The uncertainty results in more than 15% error in scaled velocity variance estimates, which are commonly used in atmospheric models. Limited comparison between microscale model $z_i$ estimates to RF model

and radiosonde $z_i$ values show increased correlation during heightened land-air interaction events. Large-eddy simulation estimates are observed to match the convective $z_i$ variability as estimated by the RF model while the global model performance is variable. Both the models do not capture the evening transitional decay of turbulence accurately.

As a part of future work, from an operational perspective improved data imputation models will be built to better ingest missing data, a RF model $z_i$ uncertainty framework and studying the effect of near-by wind farms and surface variability

across multiple sites at SGP on the boundary layer height variability using an RF model.

## Appendix A

The RF model develops several regression trees by regrouping the data based on several input features. Figure A1 below shows one of the trees developed by the model used in this article. The leaf nodes are the $z_i$ estimates from radiosondes and

the split nodes represent the surface and lidar data shown in Table 1. MATLAB built-in functions were used for the development of the RF model.





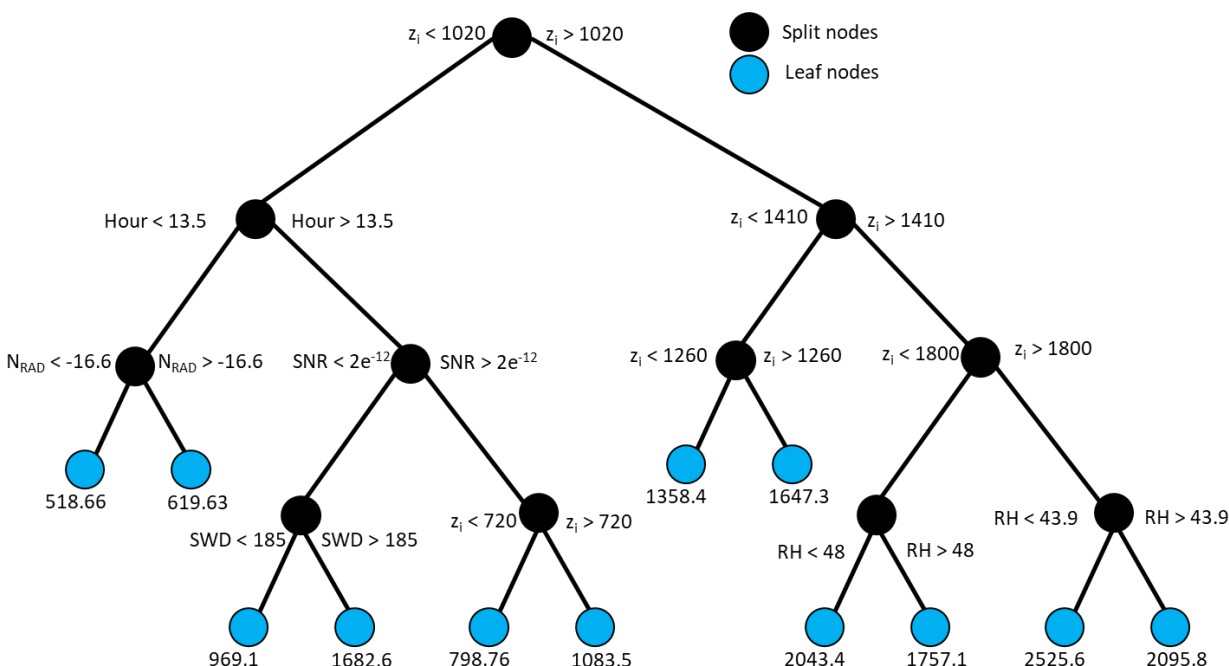

**Figure A1: A sample regression tree developed by the RF model. $Z_i$ is the Tucker Method boundary layer height estimate, SNR is signal to noise ratio, NRAD is the normal radiation, RH is relative humidity, SWD is surface wind direction, Hour is hour of the day. The Black circles represent the split nodes and the cyan circles represent the leaf nodes developed for this regression tree (aka boundary layer height estimates from radiosondes).**

**Acknowledgements**

The authors would like to acknowledge the support of Atmospheric Radiation Measurement (ARM) User Facility and the Atmospheric System Research (ASR) program for this research. The authors thank the ARM staff, instrument mentors for providing processed data, and guidance on the data archive. The authors also thank helpful discussions with Dr. Duli Chand and Dr. Joe Hardin of PNNL during model development. The Pacific Northwest National Laboratory is operated for the DOE by Battelle Memorial Institute under Contract DE-AC05-76RLO1830.

**Data Availability**

All data used in this article are publicly available data on the Atmospheric Radiation Measurement (ARM) webpage. (https://adc.arm.gov/discovery/#/). Appropriate labels and manual citations for all the data used are provided in the manuscript.

**Author Contribution**

RK and RN conceptualized the concept, RK, RN and DDT were involved in the development of the algorithm, RK did the data processing and analysis of the RF algorithm, surface stations, lidar and radiosonde results, HX performed the large-eddy simulation runs, PM performed the E3SM runs, LB calculated the boundary layer height from model outputs, RK wrote the manuscript with contributions from all authors.

**Competing interests**



The authors declare no competing interests.





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
