# Peer review of "On the estimation of boundary layer heights: A machine learning approach"

_Atmospheric Measurement Techniques, 2020_

## Referee Comment (RC1) · Anonymous Referee #1 · 18 Feb 2021

Review for AMT-2020-439 Title: "On the estimation of boundary layer heights: A machine learning approach" Authors: R. Krishnamurthy, R. K. Newsom, L. K. Berg, H. Xiao, P.-L. Ma, D. D. Turner

=================================================================

Major comments: The manuscript presents a machine learning approach (random forest) used for the computation of the boundary layer height form Doppler lidar measurements. The algorithm uses as input some parameters derived from Doppler lidars observations as well as different surface meteorological measurements. The topic is very interesting as boundary layer heights are a key parameter in many aspects. I had to re-read it several times because things were not explained clearly along the way. I am still confused on what are the variables form the Doppler lidar that are really used

as input for the RF method. I think that the manuscript will be ready for publication after the authors will address some of the concerns discussed below.

1) The main question that I have is why the authors decided to use the Tucker method for comparison and not the Bonin et al. method, which is more refined and accurate than the Tucker method. Bonin et al. (2018) presents a technique that blends all the data together from multiple scans to determine a unified measurement of the MH and the uncertainty of the estimate, but it can also be used be used with limited inputs, such as only data from zenith stares, so I am not sure why the authors did not choose it. Moreover, the clear limitation of the Tucker method (visible in Fig. 2, 3, 6, 7) is that even in the case of loss of signal, due to the low signal to noise ratio of the Doppler lidars (Fig. 6 is a clear example of this) it still provides an estimation (biased low), while the MH is clearly above that value. Fig. 3 clearly shows that the y-axes estimations (Tucker method) do not go often above 2 km, while the radiosonde ones (all three methods) have a lot of estimations above 2 km. Maybe there is a good reason for their choice, such as some limitation determined by the dataset, but I think it was not well addressed in the manuscript. As it is presented now, the RF Machine learning method seems to be used as a bias correction method applied to the Tucker method. I.e., they use the Tucker method $z_i$ as input, knowing that it is biased low when the boundary layer grows tall; they include surface observations; and they use the radiosonde estimates to train the RF method to correct it for the low bias it has during well-developed boundary layers.

2) The machine learning algorithm is somehow presented in Section 3, but in a generic context that I am not sure make it reproducible by an interested reader. I think Section 3 could be expanded and clarified in the revised version of the manuscript.

3) Another concern is that the results presented in Section 4 use a dataset to train the RF model with no missing data (which 'could' be OK), but also verify it on a dataset (future features) with no missing data, which is an ideal situation, that does not happen often in reality. Therefore, the results presented in Section 4 are on the optimistic side.

I am curious on why they just don't use the verification dataset in 2019 as is.

4) Section 6 "Case Study: Preliminary Model Comparisons" is a very, extremely preliminary test, with no quantitative results, due to the very limited number of days available to the analysis. For this reason, I am not sure I find it very useful in this study. Just my opinion.

5) Finally, I strongly suggest the authors to look for grammatical mistakes, as I found many, some of which, but not all, reported below.

===================================================================

Specific comments:

Page 1, line 12: Please, clarify the meaning of "long-term data"

Page 1, line 19: "using Doppler lidars only."

Page 1, line 19: "improvements . . . were observed"

Page 1, lines 20-21: This sentence does not read well: "where a 50% improvement in mean absolute error compared to lidar-only $z_i$ estimates and provided an R2 of greater than 85%."

Page 1, line 28: "the top of the PBL is that the turbulence is near zero". What about cloudy conditions?

Page 2, line 34: "de facto"

Page 2, line 47: The Bonin et al. (2018) reference should be placed here.

Page 2, lines 51-52: "Alternatively, velocity information from a Doppler lidar can be used to estimate $z_i$." I think this sentence is not well explained and possibly not well positioned.

Page 3, line 77: "Bianco and Wilczak 2002 and Bonin et al., 2018" are referenced to in the wrong place. They employ fuzzy logic-based methods to estimate boundary layer

heights (as stated in their titles), so, no ML used there.

Page 3, line 91: ", and data are compared". Replace with ", and observations are compared"

Page 5, Table 1: Instead of column 2 that now presents the data stream names (which could be rather included in an Appendix), I think it would be wiser to have a column introducing the height (or ranges of heights) of each measurement used in the RF machine learning method. As presented now I am confused why the vertical velocity, range-corrected attenuated backscatter, signal to noise ratio variance go from surface to 800 m AGL, as well as the average eddy dissipation rate (also why do they stop at 800m?), but no range of heights are specified for the height-resolved vertical velocity variance, wind speed and direction.

Page 6, line 114-116: What is the vertical range of measurements for the Doppler Lidar (min, max)? It is not mentioned in the text nor in Table 1. Also, it would be interesting to see the % data availability with height of the Doppler lidar used in the study.

Page 6, lines 124-125: "Estimates of eddy dissipation rate were computed between 100 to 800 m AGL". Is it only eddy dissipation rate that is computed between up to 800m? In Table 1 you mention that also all the other variables (Vertical velocity, range-corrected attenuated backscatter, signal to noise ratio (SNR) variance) are measured "from surface to 800 m AGL", but this cannot be right. Can you clarify?

Page 7, lines 155-157: It seems a large source of error the fact that even with a re-duction of sensitivity during the hottest portion of the day the algorithm still provided estimates, which are of course biased low.

Page 6, lines 129-130: Since radiosonde launches are at ∼0530, 1130, 1730 and 2330 UTC each day (local time = UTC - 0600 hours), and you are only evaluating daytime performances of your ML method, are you only using the radiosonde launches at 1730 and 2330 UTC to verify it?

[Figure]

Page 7, line 163: "radiosonde-derived zi are assumed to be the best guess zi estimate and is used to calibrate". Please, correct the grammar.

Page 8, lines 165-166: "1785 days with daytime clear and shallow cumulus conditions". What dataset are you using here? 1785 days are almost 5 years of data.

Page 8, lines 173-174: "In this paper, the primary focus is on evaluating the daytime zi estimates from RF models." How do you determine the "daytime" start and end times in your study? Do they change according to the time of the year?

Page 8, lines 183-184: "Bootstrap aggregation (bagging) is used so that each tree can randomly sample from the dataset with replacement, while only a random subset of the total feature set is given to each individual tree". I don't understand what you are doing. Could you try to rephrase this sentence? Page 10, lines 208-209: Check the grammar.

Page 10, line 226: "order of magnitude in their variability". Do you mean "in their value"?

Page 10, line 226: The word "data" is plural. Please correct here and elsewhere in the manuscript.

Page 10, lines 226-229: This whole sentence is very convoluted. Please try to rephrase it. Also, please clarify what you mean by "standardized".

Page 11, lines 256-257: "Therefore, in this analysis, the model is trained with no missing data, and no imputation is done on the data (either input or future features) to accurately test the efficacy of the RF model." This is a main concern to me. If I understand correctly, but maybe I am wrong, here you are saying that for the results presented in Section 4 you use to train and verify your model only using data with no missing features. Nonetheless, earlier in the text you stated that "It is critical for the RF model to deal with missing values in its training phase", which I strongly agree with, as in real life missing features can happen. If this is true (again, I might have misunderstood), I think your results are more representative of a best-case real-time scenario, without

ever missing features.

Page 11, line 262: "where in a model developed at a given site is tested", "in" can be removed.

Page 12, line 283: I see that you define "daytime" here. Maybe you could specify it earlier in the text, when you first talk about it.

Page 13, line 307: "with an R2 of greater than 0.85", remove "of"

Page 13, line 315: "estimates that used to calibrate the model", replace with "estimates that are used to calibrate the model".

Page 13, line 320: Please reword "2 annual cycles of data"?

Page 14, Figure caption: Please specify that c) and d) are for "daytime and nighttime".

Pages 15, Fig. 6: It seems that the RF method can provide estimated where there are no Doppler lidar measurements. Is that correct?

Pages 15 and 16, Figs. 6 and 7: Please keep the colors the same in both figures (i.e.: Lidar zi should be red in Fig. 7).

Page 16, line 359: "are shown in Figure 8a"

Page 16, Fig. 7 caption: I think here you could simply say "As in Fig. 6, but for June 22, 2019."

Page 16, line 363: "a standard bias correction would not always improve zi estimates from the Tucker method". Actually, the bias in the Tucker method seems pretty constant in Fig. 8a. . . Which is confusing because before you said that "the Tucker method generally works well at tracking the height of the mixed layer during its initial development phase" and this does not reflect in the gray line in Fig. 8a.

Page 17, Figure caption: You say that in panel a) there is the "Tucker method zi", but you call it "Lidar-only zi" in the label. Sometimes you refer to it as the "Lidar-only"

method and sometime as the "Tucker method", here and in other places in the text. Also, how is the "Lidar-only zi" determined at nigh-time? Here it seems that it cannot be the lowest range-gate of the Doppler lidar as it seems higher than the one in Fig. 6. Also, please, specify what the error bars represent.

Pages 18 and 19, Tables 4 and 5. Very interesting analysis and results. Would it be possible in your future research to include land use type? Could this give you the possibility to include a variable to distinguish between different sites/seasons in your future analysis?

Page 19, line 419: This should be Eq. (4), not (1).

Page 19, Table 5: Is this "Lidar-only zi" simply the lowest range-gate of the Doppler lidar?

Page 20, line 442: "that the parameters shown to be important are with respect to the RF model, are features that successfully...". Remove one "are".

Page 20, lines 447-448: "In this research, we have mostly analysed using standard processed data from SGP instruments as an input into the RF model." This sentence is incomplete.

Page 21. Fig. 9: Should y-label include (%)?

Page 22: The Equation numbers are wrong.

Page 23, line 496: What are "lidar false alarm rates"?

Pages 24 and 25: I find this whole section not very interesting. The models are described in great detail, but the analysis is very poor, due to the very limited dataset. So, I don't know if it adds much to the manuscript.

Page 24, line 517: "horizontal resolution". Would it be more accurate to say, "horizontal grid spacing"?

Page 24, lines 530-531: "Nocturnal $z_i$ is not estimated using model data". Why is that?

Page 24, line 535: "RF model provides $z_i$ estimates at a much finer temporal resolution than radiosondes". What is the RF model temporal resolution?

Page 26, line 566: "the mean absolute error of boundary layer height estimated by RF model reduced", "is" is missing.

Page 26, lines 577-578: "in convective velocity scale estimates when used Tucker method." Replace with "in convective velocity scale estimates when the Tucker method is used."

Page 26, lines 583-585: The whole sentence is poorly written, please rephrase.

---

## Referee Comment (RC2) · Anonymous Referee #2 · 3 Mar 2021

Review of the article titled "On the estimation of boundary layer heights: a machine learning approach" by Krishnamurthy and coauthors for publication in the Atmospheric Measurement Technique.

The authors have used a machine learning (ML) approach to improve the retrieval of boundary layer depth from the data collected by the Doppler Lidar. They first develop a ML model to calibrate the DL retrieved PBL depth with that derived from the radiosondes. As the radiosonde measurements are temporally sparse, they use the higher resolution PBL depth retrieved from the Doppler Lidar to understand boundary layer parameters affecting it. In the end they also evaluate two days of output from two different models. The article is overall well-written and is easy to follow. However, the article can be further improved by addressing the following concerns. These can be

regarded as minor revisions.

Major Concerns:

It will be good to add some discussion in the last section on the use of machine learning in deducing PBL depth, and understanding its controls. The authors have mentioned and acknowledged several things in the text, i) like the training could have been performed by using a different estimate of PBL depth from the radiosonde, and ii) how the authors are only demonstrating the use of ML for deriving the PBL in the nighttime, but refrain to call it the "true" nighttime $Z_i$ (Line 313-315). This is simple the limitation of the use of ML in deriving physical understanding. This should be discussed in the text in detail. If the authors truly believe (#2 above) to be the case, then can you trust the numbers reported in Table 4 and 5? Maybe the Tucker method is correct and just the training needs to be done on a different dataset. This concern does not mean that the article is not valuable, however this needs to be addressed in the text. Thank you.

Figure 10 and associated text: it is a bit confusing as to the whole purpose of this exercise. Just because the variance is being scaled by a higher PBL depth, the profile will look different. So not sure how it speaks to the Random Forest (RF) PBL depth being better than that derived by the Tucker method. Also, the variability of variance is probably huge, so the differences wouldn't be statistically significant anyways. This needs to be clarified in the text, or else removed from the manuscript. Thanks.

Minor Concerns:

Line 14: Might be better to say four years rather than multi-year. Thanks.

Line 41: MISR is mis-spelled.

Line 42:43: The satellites measure cloud top temperature from which the cloud top heights are calculated. During cloudy conditions, it is assumed that the PBL top corresponds to cloud top heights. This statement states that there has not been any validation of the satellite derived cloud top heights. Please add reference to support

this, or else remove. Thanks.

Line 58: you mean rely and not relay.

Line 64: better word would be "lowest gate" rather than minimum range.

Table 1: It will be good if you add units to the measurement features. Thanks.

Line 145-146: please revise this sentence. Thanks.

Line 165: The numbers do not add up. Four years of data should equal 1460 days, not sure how you got 1785 days.

Line 285: you mean "hourly" cloud fraction greater than 0.1?

Figure 8: Please describe the vertical bars in the caption.

Figure 11: Looks like the LASSO simulations are able to accurately capture the development of the daytime PBL. I assume that the E3SM values are within the model range resolution as well. So this is very good news for the modelling community and should be highlighted.

---

## Author Comment (AC1) · 31 Mar 2021

Review for AMT-2020-439 Title: "On the estimation of boundary layer heights: A machine learning approach"
Authors: R. Krishnamurthy, R. K. Newsom, L. K. Berg, H. Xiao, P.-L. Ma, D. D. Turner
==============================================================================================
Major comments: The manuscript presents a machine learning approach (random forest) used for the computation of the boundary layer height form Doppler lidar measurements. The algorithm uses as input some parameters derived from Doppler lidars observations as well as different surface meteorological measurements. The topic is very interesting as boundary layer heights are a key parameter in many aspects. I had to re-read it several times because things were not explained clearly along the way. I am still confused on what are the variables form the Doppler lidar that are really used C1 as input for the RF method. I think that the manuscript will be ready for publication after the authors will address some of the concerns discussed below.

We thank the reviewer for carefully reading the article and providing constructive feedback. We believe the quality of the article has improved by addressing the comments and hope our revisions are acceptable to the reviewer. Below the reviewer comments are in black and the authors responses are in blue.

1) The main question that I have is why the authors decided to use the Tucker method for comparison and not the Bonin et al. method, which is more refined and accurate than the Tucker method. Bonin et al. (2018) presents a technique that blends all the data together from multiple scans to determine a unified measurement of the MH and the uncertainty of the estimate, but it can also be used be used with limited inputs, such as only data from zenith stares, so I am not sure why the authors did not choose it. Moreover, the clear limitation of the Tucker method (visible in Fig. 2, 3, 6, 7) is that even in the case of loss of signal, due to the low signal to noise ratio of the Doppler lidars (Fig. 6 is a clear example of this) it still provides an estimation (biased low), while the MH is clearly above that value. Fig. 3 clearly shows that the y-axes estimations (Tucker method) do not go often above 2 km, while the radiosonde ones (all three methods) have a lot of estimations above 2 km. Maybe there is a good reason for their choice, such as some limitation determined by the dataset, but I think it was not well addressed in the manuscript. As it is presented now, the RF Machine learning method seems to be used as a bias correction method applied to the Tucker method. I.e., they use the Tucker method $z_i$ as input, knowing that it is biased low when the boundary layer grows tall; they include surface observations; and they use the radiosonde estimates to train the RF method to correct it for the low bias it has during well-developed boundary layers.

At all the ARM SGP sites, the coherent Doppler lidars are programmed to perform vertical stares with an 8-point Velocity Azimuth Display (VAD) scan once every 15 minutes. Good data is generally only available for heights much less than 2 km (Newsom and Krishnamurthy 2020) and we have provided the data availability of the Doppler lidar vertical stares and VAD scans for the study period in Figures below. We have also included these in Appendix A of the updated manuscript. With poor SNR returns at higher altitudes, the dataset is generally unreliable and has a high false alarm rate (explained in our response to another question raised by the reviewer below). This would create an additional level of complexity in using other algorithms on the SGP dataset. The Tucker method is a standard methodology and has been used by many to estimate the boundary layer height at the SGP site an other regions (Träumner et al., 2011, Shukla et al., 2014, Schween et al., 2014, Berg et al., 2017, Lareau et al., 2018, Lareau 2020), and given the above limitations of the ARM dataset (only vertical stares and limited range), ease in application to the multi-year dataset, and known performance at SGP C1, the Tucker method was used as a baseline algorithm in this study for comparing with the RF model output. This also provides much needed guidance for others (who traditional use the Tucker method) to use an alternative algorithm for their research and highlight the Tucker method's deficiencies. Algorithms based on Fuzzy Logic or Wavelet transforms are indeed very interesting and we hope to do a comparative analysis in a later study using techniques like the Bonin et al. (2018) method.

The goal of the approach in this paper is to use multiple atmospheric parameters (using both remote sensing and in-situ) to infer the boundary layer height using a Random Forest model. We believe the RF model here does more than bias correct the Tucker method estimates of $z_i$ (as shown in Figure 6 and 7, the RF model matches the Tucker method during some time periods as well, showing it is not just a bias correction). It provides a higher weight to the Tucker method $z_i$ estimate compared to other observations during convective boundary layers as

shown in Table 4 of the manuscript, since the Tucker method is close to the true $z_i$. But also, equally uses other surface-based measurements to dynamically adjust the $z_i$ (See Table 4 & 5 of the manuscript – for e.g. surface temperature, TKE, friction velocity etc.,) during both convective and nocturnal conditions.

We have expanded the discussion of the baseline method in the updated manuscript, added the range plots in Appendix A (please refer to Page 7, Line 154-161)

[Figure]

Figure. ARM SGP C1 Doppler lidar range availability from vertical stares from 2016 to 2019 after signal to noise ratio filtering (ARM standard threshold of 1.008 dB).

[Figure]

Figure. ARM SGP C1 Doppler lidar wind speed profile range availability from processed VAD scans from 2016 to 2019 after signal to noise ratio filtering (ARM standard threshold of 1.008 dB).

References:

Träumner, K., Kottmeier, C., Corsmeier, U., & Wieser, A. (2011). Convective boundary-layer entrainment: Short review and progress using Doppler lidar. Boundary-layer meteorology, 141(3), 369-391.

Shukla, K. K., Phanikumar, D. V., Newsom, R. K., Kumar, K. N., Ratnam, M. V., Naja, M., & Singh, N. (2014). Estimation of the mixing layer height over a high altitude site in Central Himalayan region by using Doppler lidar. Journal of Atmospheric and Solar-Terrestrial Physics, 109, 48-53.

Schween, J. H., Hirsikko, A., Löhnert, U., & Crewell, S. (2014). Mixing-layer height retrieval with ceilometer and Doppler lidar: from case studies to long-term assessment. Atmospheric Measurement Techniques, 7(11), 3685-3704.

Lareau, N. P., Zhang, Y., & Klein, S. A. (2018). Observed boundary layer controls on shallow cumulus at the ARM Southern Great Plains site. Journal of the Atmospheric Sciences, 75(7), 2235-2255.

Lareau, N. P. (2020). Subcloud and Cloud-Base Latent Heat Fluxes during Shallow Cumulus Convection. Journal of the Atmospheric Sciences, 77(3), 1081-1100.

2) The machine learning algorithm is somehow presented in Section 3, but in a generic context that I am not sure make it reproducible by an interested reader. I think Section 3 could be expanded and clarified in the revised version of the manuscript.

Thank you for the comment, we have included a brief description of the process and have included the MATLAB packages and commands that were used in building the RF model.  Please see Section 3.1 and Page 11 line 262 to 267 of the updated manuscript for additional details.  Implementing this algorithm in MATLAB given the functions used in this study, would aide the reader in quickly implementing this algorithm and the references within MATLAB to understand basic Random Forest framework.

3) Another concern is that the results presented in Section 4 use a dataset to train the RF model with no missing data (which 'could' be OK), but also verify it on a dataset (future features) with no missing data, which is an ideal situation, that does not happen often in reality. Therefore, the results presented in Section 4 are on the optimistic side. I am curious on why they just don't use the verification dataset in 2019 as is.

This is a very good and valid point raised by the reviewer. We agree that the results presented in Section 4 are on the optimistic side. We would like to first note that the issue of treating missing data is a larger topic of research while using machine learning models (please see Hastie et al., 2008, Tang and Ishwaran 2017). Assessing the performance of the RF algorithm using data imputation creates an additional level of complexity when one of the features is missing for a given time step compared to others. For example, if Relative Humidity (an important variable for $z_i$ estimation as shown in Table 4) data is a missing feature for a given time step, how does that change the accuracy of the RF output compared to when the surface temperature (also an important variable for $z_i$ estimation) data is missing for a given time step? There are several possible scenarios here (as we have close to 20 odd features), and we won't be capturing all the associated errors effectively, as each of the features are not weighted equally in the model. In addition to that, when the data is missing, the RF algorithm does data imputation based on neighboring available time series data (see Section 3.2). This imputation process adds further uncertainty to the features, and it is not trivial to assess the effect of data imputation on one of the variables over the entire output (as it depends on the duration of the data loss as well). Therefore, we decided not to evaluate RF model outputs with missing features but we do show the reader that this is an important aspect that one needs to consider when using this algorithm. Several imputation algorithms are available in the literature and we are currently working on a few imputation algorithms to test the accuracy of the output at SGP, and hope to advance on this topic in our research. We have added further explanation on this topic in Section 3 (Page 12 Line 294 to 300).

Reference:
        Hastie, T., Tibshirani, R., and Friedman, J., 2009. The elements of statistical learning: data mining, inference, and prediction. Springer Science & Business Media.
        Tang, F., & Ishwaran, H. (2017). Random forest missing data algorithms. Statistical Analysis and Data Mining: The ASA Data Science Journal, 10(6), 363-377.

4) Section 6 "Case Study: Preliminary Model Comparisons" is a very, extremely preliminary test, with no quantitative results, due to the very limited number of days available to the analysis. For this reason, I am not sure I find it very useful in this study. Just my opinion.

As the reviewer probably is aware, Large-eddy simulation (LES) and E3SM models are expensive to run and LES models especially can only be run for short durations and for specific cases. This indeed limits our ability to provide quantitative results in validating models and observations. Since we hope to create an Atmospheric Radiation Measurement (ARM) Value-Added Product (VAP) using the RF model to estimate the boundary layer height, we believe this preliminary inter-comparison is important. This section is a case-study and shows the modeling community a framework that can be used to evaluate climate and LES models and how this type of VAP products can help us improve our understanding of the PBL properties. We expect future models to use this VAP for validating boundary layer height and its impact on model physics over much longer time periods. We have added further our motivation for the case study in Section 6.

5) Finally, I strongly suggest the authors to look for grammatical mistakes, as I found many, some of which, but not all, reported below.

Thank you for your careful review, we very much appreciate it. We have made changes to text, where we felt grammatical corrections were necessary. The article was also subsequently reviewed by a technical editor at PNNL.

Specific comments:
Page 1, line 12: Please, clarify the meaning of "long-term data"

By "long-term data", we mean continuous measurements for several years at a given site, for example the ARM data at SGP C1 has been collecting data for more than 3 decades. We have changed the wording to "...multi-year dataset."

Page 1, line 19: "using Doppler lidars only."

Changed.

Page 1, line 19: "improvements : : : were observed"

Changed.

Page 1, lines 20-21: This sentence does not read well: "where a 50% improvement in mean absolute error compared to lidar-only $z_i$ estimates and provided an R2 of greater than 85%."

We have rephrased the sentence to: "Noteworthy improvements in daytime $z_i$ estimates were observed using the RF model, with a 50% improvement in mean absolute error and an $R^2$ of greater than 85% compared to the Tucker method $z_i$."

Page 1, line 28: "the top of the PBL is that the turbulence is near zero". What about cloudy conditions?

We agree that during cloudy periods, considerable turbulence can be observed at the cloud base height. We have amended this statement.
"One of the characteristics of the top of the PBL during clear sky conditions is that the turbulence is near zero, while during cloudy conditions, significant updrafts and downdrafts can be observed at the top of the boundary layer (typically the cloud base height). "

Page 2, line 34: "de facto"

Corrected.

Page 2, line 47: The Bonin et al. (2018) reference should be placed here.

Agreed and corrected.

Page 2, lines 51-52: "Alternatively, velocity information from a Doppler lidar can be used to estimate zi." I think this sentence is not well explained and possibly not well positioned.

We have amended the sentence and moved it to the Page 3 line 66 and provided additional references.
"Horizontal velocity variance and dissipation rate profiles from a Doppler lidar can be used to estimate $z_i$ in nocturnal conditions (Vakkari et al., 2015, Banakh et al., 2021)."

Vakkari, V., O'Connor, E. J., Nisantzi, A., Mamouri, R. E., & Hadjimitsis, D. G. (2015). Low-level mixing height detection in coastal locations with a scanning Doppler lidar. Atmospheric Measurement Techniques, 8(4), 1875-1885.
Banakh, V. A., Smalikho, I. N., & Falits, A. V. (2021). Estimation of the height of the turbulent mixing layer from data of Doppler lidar measurements using conical scanning by a probe beam. Atmospheric Measurement Techniques, 14(2), 1511-1524.

Page 3, line 77: "Bianco and Wilczak 2002 and Bonin et al., 2018" are referenced to in the wrong place. They employ fuzzy logic-based methods to estimate boundary layer heights (as stated in their titles), so, no ML used there.

Agreed, this was an oversight on our end. We have referenced Bonin et al., 2018 as mentioned earlier.

Page 3, line 91: ", and data are compared". Replace with ", and observations are compared"

Changed.

Page 5, Table 1: Instead of column 2 that now presents the data stream names (which could be rather included in an Appendix), I think it would be wiser to have a column introducing the height (or ranges of heights) of each measurement used in the RF machine learning method. As presented now I am confused why the vertical velocity, range-corrected attenuated backscatter, signal to noise ratio variance go from surface to 800 m AGL, as well as the average eddy dissipation rate (also why do they stop at 800m?), but no range of heights are specified for the height-resolved vertical velocity variance, wind speed and direction.

We have modified the table considerably, as it contained a mix of features that are being used in the model and data that are used to derive certain features. We have now limited the table to only include features used in the RF model. We have also included the height/range of measurement and units of various features.

Due to the limited range of the Doppler lidars at SGP, estimates of eddy dissipation rate were significantly affected by noise of the system at higher altitudes. Therefore, measurements are restricted to the highest signal-to-noise ratio portion of the profile. We observed that estimates below 800 m were reasonably stable for extended time periods and hence limited our dataset to 800 m AGL. Similarly, we also limited other variables from the Doppler lidar to 800 m AGL. The reason for taking the spatial average of the features, is the RF algorithm cannot ingest 2-Dimensional data (time x height) and can typically only use timeseries data at input features. Datasets at specific heights can be provided as time series input. Therefore, we decided to take either the variance or average of the features to capture some of the atmospheric dynamics within the lowest 800 m. Adding these features increased the overall variance explained by the RF model (82% of the total variance). We have added this explanation in Section 2 of the updated manuscript.

Page 6, line 114-116: What is the vertical range of measurements for the Doppler Lidar (min, max)? It is not mentioned in the text nor in Table 1. Also, it would be interesting to see the % data availability with height of the Doppler lidar used in the study.

The availability of the vertical stares from the ARM Doppler lidars at SGP C1 for both vertical stares and VAD scans is shown again below and further details are provided in the ARM Doppler lidar handbook (Newsom and Krishnamurthy 2020). We have included these in Appendix A of the updated manuscript. We would also refer readers to the handbook for additional details on the ARM Doppler lidars.

[Figure]

Figure. ARM SGP C1 Doppler lidar range availability from vertical stare scans from 2016 to 2019 after signal to noise ratio filtering (threshold of 1.008 dB).

[Figure]

Figure. ARM SGP C1 Doppler lidar wind speed profile range availability from processed VAD scans from 2016 to 2019 after signal to noise ratio filtering (threshold of 1.008 dB).

Page 6, lines 124-125: "Estimates of eddy dissipation rate were computed between 100 to 800 m AGL". Is it only eddy dissipation rate that is computed between up to 800m? In Table 1 you mention that also all the other variables (Vertical velocity, range corrected attenuated backscatter, signal to noise ratio (SNR) variance) are measured "from surface to 800 m AGL", but this cannot be right. Can you clarify?

Addressed in the comment above.  This has been further clarified in the updated manuscript (Page 6 line 127-133).

Page 7, lines 155-157: It seems a large source of error the fact that even with a reduction of sensitivity during the hottest portion of the day the algorithm still provided estimates, which are of course biased low.

Yes, the Tucker method does provide reasonable estimates of $z_i$ but is limited by the ability of the Doppler lidar to potently see up to the boundary layer during strongly convective conditions with deep boundary layers as one would generally observe small amounts of aerosol loading near the top of the boundary layer.  As shown in Table 4 of the manuscript, the Tucker method is weighted heavily in the RF model due to its proximity to true $z_i$.  Therefore, if there were a powerful Doppler lidar at SGP (say the Lockheed Martin WindTracer or similar), and the lidar could measure up to the boundary layer height and above during low aerosol loading conditions (clear air conditions might still be an issue), the RF model $z_i$ will match the Tucker method $z_i$.

Page 6, lines 129-130: Since radiosonde launches are at _0530, 1130, 1730 and 2330 UTC each day (local time = UTC - 0600 hours), and you are only evaluating daytime performances of your ML method, are you only using the radiosonde launches at 1730 and 2330 UTC to verify it?

We are evaluating the RF method using all the radiosondes deployed at SGP C1 in this article.  The nocturnal conditions are typically evaluated using the 0530- and 1130-hour radiosondes and convective conditions using 1730- and 2330-hour radiosondes.  During select field campaigns, additional radiosondes are deployed with a higher temporal frequency which are also used in our evaluation (Figure 6 & 7).

Page 7, line 163: "radiosonde-derived zi are assumed to be the best guess zi estimate and is used to calibrate". Please, correct the grammar.

We have rephrased it to : "…radiosonde-derived $z_i$ estimates are used to calibrate the RF algorithm".

Page 8, lines 165-166: "1785 days with daytime clear and shallow cumulus conditions". What dataset are you using here? 1785 days are almost 5 years of data.

The ARM site has at least 2 radiosondes during daytime conditions and a higher frequency during some field campaigns, so for 4 years there are approximately 2920 radiosonde launches. We used 4 years of data and found approximately 1785 cases (not days) in our analysis. This typo was fixed and the statement has been rephrased to "…1785 cases with radiosonde data and daytime clear (identified as periods when surface heat flux is positive from sunrise to sunset and cloud base height is zero) or shallow cumulus conditions (identified as cloud base height less than 5 km from Doppler lidar and cloud fraction less than 0.1) for the years 2016 through 2019."

Page 8, lines 173-174: "In this paper, the primary focus is on evaluating the daytime zi estimates from RF models." How do you determine the "daytime" start and end times in your study? Do they change according to the time of the year?

We agree with the reviewer, we had mentioned how we delineate daytime conditions in the results section but have now provided a note on how we pick daytime conditions when we first mention daytime.

Page 8, lines 183-184: "Bootstrap aggregation (bagging) is used so that each tree can randomly sample from the dataset with replacement, while only a random subset of the total feature set is given to each individual tree". I don't understand what you are doing. Could you try to rephrase this sentence? Page 10, lines 208-209: Check the grammar.

We have rephrased it to: "Bootstrap aggregation (bagging) is used so that each RF tree (sample shown in Appendix B) can randomly sample from an entire feature set, while only a subset of the total feature set is given to each individual tree. For example, if the entire feature set contains say N different features, an individual RF tree can contain a fraction of those N features."

Page 10, line 226: "order of magnitude in their variability". Do you mean "in their value"?

Correct, we have rephrased it to "order of magnitude in their value".

Page 10, line 226: The word "data" is plural. Please correct here and elsewhere in the manuscript.

Yes, we have now corrected for this in the updated manuscript.

Page 10, lines 226-229: This whole sentence is very convoluted. Please try to rephrase it. Also, please clarify what you mean by "standardized".

We have rephrased it to read: "Standardizing involves aligning the features to have a zero mean and scaled to have standard deviation of one. Typically, RF models do not need standardizing or normalizing features due to the inherent bagging process (Breiman, 2001). At SGP C1, large diurnal variability was observed for certain parameters (like TKE, dissipation rate, etc.), leading to a large distribution of values between daytime and nighttime. This

skewed the number of trees the RF model builds for daytime and nighttime estimates. Standardizing the features showed improvements while estimating nocturnal $z_i$ estimates from the RF model."

Page 11, lines 256-257: "Therefore, in this analysis, the model is trained with no missing data, and no imputation is done on the data (either input or future features) to accurately test the efficacy of the RF model." This is a main concern to me. If I understand correctly, but maybe I am wrong, here you are saying that for the results presented in Section 4 you use to train and verify your model only using data with no missing features. Nonetheless, earlier in the text you stated that "It is critical for the RF model to deal with missing values in its training phase", which I strongly agree with, as in real life missing features can happen. If this is true (again, I might have misunderstood), I think your results are more representative of a best-case real-time scenario, without ever missing features.

The reviewer has not misunderstood, and her/his feelings about the results are correctly placed. We have provided explanation to this issue in Major comments #3. We hope that provides sufficient detail for the reviewer to understand the reasoning behind our choice.

Page 11, line 262: "where in a model developed at a given site is tested", "in" can be removed.

Corrected.

Page 12, line 283: I see that you define "daytime" here. Maybe you could specify it earlier in the text, when you first talk about it.

Agreed, we have mentioned it earlier in the updated manuscript as well.

Page 13, line 307: "with an R2 of greater than 0.85", remove "of"
Corrected.

Page 13, line 315: "estimates that used to calibrate the model", replace with "estimates that are used to calibrate the model".

Changed.

Page 13, line 320: Please reword "2 annual cycles of data"?

It has been rephrased to: "…the correlation coefficient was negligible using at least two years of data."

Page 14, Figure caption: Please specify that c) and d) are for "daytime and nighttime".

Yes, it has been changed.

Pages 15, Fig. 6: It seems that the RF method can provide estimated where there are no Doppler lidar measurements. Is that correct?

In Figure 6, there are no future missing features, so there are no missing Doppler lidar measurements. But yes, the reviewer is technically correct when we do the processing using missing future features the RF approach could provide estimates of $z_i$ when data from the Doppler lidar is missing. If the reviewer is referring to estimating boundary layer height above the lidar range during convective boundary layers in Figure 6, that is not considered missing in our study as we do not give the 2-dimensional data as input to the RF model.

Pages 15 and 16, Figs. 6 and 7: Please keep the colors the same in both figures (i.e.: Lidar zi should be red in Fig. 7).

Thanks for noticing that, the colors are now consistent in both figures.

Page 16, line 359: "are shown in Figure 8a"

Changed.

Page 16, Fig. 7 caption: I think here you could simply say "As in Fig. 6, but for June 22, 2019."

Changed.

Page 16, line 363: "a standard bias correction would not always improve zi estimates from the Tucker method". Actually, the bias in the Tucker method seems pretty constant in Fig. 8a: : : Which is confusing because before you said that "the Tucker method generally works well at tracking the height of the mixed layer during its initial development phase" and this does not reflect in the gray line in Fig. 8a.

We have rephrased the sentence in Page 7, Line 167 to read: "We find that the Tucker method generally works well at tracking the height of the convective mixed layer during its initial development phase and can match the radiosonde observations by changing the vertical velocity variance threshold (Schween et al., 2014)."

The choice of the threshold is somewhat subjective, and by using a lower threshold we can match the radiosonde observations during the initial development phase. We use a standard 0.04 $m^2s^{-2}$ as provided in Tucker et al., 2009 paper. There have been a few studies showing that varying this threshold can increase the correlation with radiosonde measurements (Schween et al., 2014). But for RF approach used in this article, investigating the choice of the threshold is somewhat irrelevant, since we are using the data from the Doppler lidars as a first guess. Therefore, we don't think that refining this threshold would significantly alter our results.

Schween, J. H., Hirsikko, A., Löhnert, U., & Crewell, S. (2014). Mixing-layer height retrieval with ceilometer and Doppler lidar: from case studies to long-term assessment. Atmospheric Measurement Techniques, 7(11), 3685-3704.

Page 17, Figure caption: You say that in panel a) there is the "Tucker method zi", but you call it "Lidar-only zi" in the label. Sometimes you refer to it as the "Lidar-only" method and sometime as the "Tucker method", here and in other places in the text. Also, how is the "Lidar-only zi" determined at nigh-time? Here it seems that it cannot be the lowest range-gate of the Doppler lidar as it seems higher than the one in Fig. 6. Also, please, specify what the error bars represent.

We have changed all such instances in the manuscript to read Tucker method to ensure consistency. We have also mentioned that the error bars represent one standard deviation in Figure 8a and 8b.

The night-time $z_i$ estimates from Tucker method are generally the lowest Doppler lidar range-gate, as the vertical velocity variance estimates during nighttime conditions are much smaller than the threshold used for this study. During certain rare nighttime periods, say convection initiation events, larger vertical velocity variance estimates are observed, and we chose to use the same threshold to provide a boundary layer height estimate during such conditions. We have added a sentence in the updated manuscript explaining the above (Page 6, line 125-129).

Pages 18 and 19, Tables 4 and 5. Very interesting analysis and results. Would it be possible in your future research to include land use type? Could this give you the possibility to include a variable to distinguish between different sites/seasons in your future analysis?

We appreciate the reviewer's comments. The RF model is typically site specific, so extending it to other sites would either need radiosondes launched at those sites for validation purposes or would require building a new model with another reference measurement for calibration. Radiosondes are only launched at SGP C1. We extended the model developed at SGP C1 to neighboring four ARM satellite sites (approximately 50 kilometers away from SGP C1), but we need additional reference $z_i$ measurements to truly understand the errors at those sites. In theory, we agree with the reviewer, if we can train a model at each of those sites, we can understand the effect of land-use type on the boundary-layer height. We have begun investigating the use of satellite datasets at these sites (cloud top height, land-use type, vegetation index, albedo etc.), and hope to provide some insight on the effect of land-use type on boundary layer height at SGP as part of our future work.

Page 19, line 419: This should be Eq. (4), not (1).

Changed.

Page 19, Table 5: Is this "Lidar-only zi" simply the lowest range-gate of the Doppler lidar?

For consistency and simplicity, we have defined Tucker Method as when the vertical velocity variance is below a certain threshold. Therefore, have changed this and the associated text to Tucker Method. We provide further explanation in Page 6 Line 125-129.

Page 20, line 442: "that the parameters shown to be important are with respect to the RF model, are features that successfully: : :". Remove one "are".

Removed one "are". Updated sentence: "It is important to note, that the parameters shown to be important with respect to the RF model, are features that successfully aide in the RF bagging process"

Page 20, lines 447-448: "In this research, we have mostly analysed using standard processed data from SGP instruments as an input into the RF model." This sentence is incomplete.

We have rephrased this to: "In this research, the input features into the RF model are standard atmospheric parameters (such as wind speed, temperature, etc.)."

Page 21. Fig. 9: Should y-label include (%)?

Partial dependence estimates are unitless. Partial dependence represents the relationships between predictor variables and predicted responses in a trained regression model (Friedman 2001). Higher the partial dependence value, larger the impact of the feature on the predicted response.

      Friedman, Jerome. H. "Greedy Function Approximation: A Gradient Boosting Machine." The Annals of Statistics 29, no. 5 (2001): 1189-1232.

Page 22: The Equation numbers are wrong.

Thanks for catching the typo, they have been corrected and are in order.

Page 23, line 496: What are "lidar false alarm rates"?

The signal to noise ratio or Carrier Noise Ratio filtering thresholds are based on False Alarm Rates (FAR, generally a FAR of 0.25% is used in the standard signal processing). FAR is defined as the probability of a Noise Peak detected in the lidar spectra that leads to retrieving a false wind speed estimate. Probability of Detection (POD), i.e., probability of detecting an accurate Doppler peak from the Lidar spectra, is set to 99.75% (Bouquet et al., 2016). Therefore, even after we filter using a standard SNR threshold, we end up with significant noise at higher range-gates. We have added the reference to the updated manuscript.

Boquet, M., Royer, P., Cariou, J. P., Machta, M., & Valla, M. (2016). Simulation of Doppler lidar measurement range and data availability. Journal of Atmospheric and Oceanic Technology, 33(5), 977-987.

Pages 24 and 25: I find this whole section not very interesting. The models are described in great detail, but the analysis is very poor, due to the very limited dataset. So, I don't know if it adds much to the manuscript.

Thank you for the comment. We feel from a perspective of future use of the RF model zi as a value-added product (VAP), this preliminary comparison paves the way for potential users to use this dataset for their validation purposes. We also observe that during clear sky conditions, the models do not accurately capture the evening transition decay of turbulence. Of course, additional analysis is needed but such systematic differences between the model and data are crucial for targeting future research directions. We have added some additional text to highlight this aspect.

Page 24, line 517: "horizontal resolution". Would it be more accurate to say, "horizontal grid spacing"?

Agreed, changed to horizontal grid spacing.

Page 24, lines 530-531: "Nocturnal zi is not estimated using model data". Why is that?

Estimates of boundary-layer height from E3SM and LASSO were not made during nocturnal conditions. The LASSO simulations extend only from approximately sunrise to sunset so nocturnal estimates of the boundary-layer height are not possible. The E3SM simulations generally have too much nighttime turbulence, making estimates of the boundary-layer height unreliable at night. We have added this to the updated manuscript.

Page 24, line 535: "RF model provides zi estimates at a much finer temporal resolution than radiosondes". What is the RF model temporal resolution?

The RF model temporal resolution is 15 minutes. We have now mentioned that in the manuscript.

Page 26, line 566: "the mean absolute error of boundary layer height estimated by RF model reduced", "is" is missing.

Added "is" to the sentence. Thanks.

Page 26, lines 577-578: "in convective velocity scale estimates when used Tucker method." Replace with "in convective velocity scale estimates when the Tucker method is used."

Replaced with the above sentence. Thanks.

Page 26, lines 583-585: The whole sentence is poorly written, please rephrase.

Rephrased to: There are a number of ways to expand on the research presented here. Future will could focus on improved data imputation models to better handle missing data, a RF model zi uncertainty framework using individual RF tree predictions, and finally a study of the effect of near-by wind farms and surface heterogeneity have on the boundary layer height.

---

## Author Comment (AC2) · 31 Mar 2021

Review of the article titled "On the estimation of boundary layer heights: a machine learning approach" by Krishnamurthy and coauthors for publication in the Atmospheric Measurement Technique.

The authors have used a machine learning (ML) approach to improve the retrieval of boundary layer depth from the data collected by the Doppler Lidar. They first develop a ML model to calibrate the DL retrieved PBL depth with that derived from the radiosondes. As the radiosonde measurements are temporally sparse, they use the higher resolution PBL depth retrieved from the Doppler Lidar to understand boundary layer parameters affecting it. In the end they also evaluate two days of output from two different models. The article is overall well-written and is easy to follow. However, the article can be further improved by addressing the following concerns. These can be regarded as minor revisions.

We thank the reviewer for carefully reading the article and providing constructive feedback. We believe the quality of the article has improved by addressing the comments and hope our revisions are acceptable to the reviewer. Below the reviewer comments are in **black** and the authors responses are in blue.

**Major Concerns:**

It will be good to add some discussion in the last section on the use of machine learning in deducing PBL depth, and understanding its controls. The authors have mentioned and acknowledged several things in the text, i) like the training could have been performed by using a different estimate of PBL depth from the radiosonde, and ii) how the authors are only demonstrating the use of ML for deriving the PBL in the nighttime, but refrain to call it the "true" nighttime Zi (Line 313-315). This is simple the limitation of the use of ML in deriving physical understanding. This should be discussed in the text in detail. If the authors truly believe (#2 above) to be the case, then can you trust the numbers reported in Table 4 and 5? Maybe the Tucker method is correct and just the training needs to be done on a different dataset. This concern does not mean that the article is not valuable, however this needs to be addressed in the text. Thank you.

These are very good points raised by the reviewer and we have tried to address these in the updated manuscript and provide further explanations below.

The sensitivity of the RF model indeed depends on the reference data used for training, but the choice was made by comparing the standard Tucker method to all radiosonde  $z_i$  estimates generally used by the community. We found that the correlations were higher based on the Tucker method, which we know is generally biased low in many conditions but is precise. So, we believe the choice of the reference data (Liu & Liang based approach) is better suited for the ARM SGP C1 site. For another site, it would be prudent to conduct a preliminary comparison against lidar derived estimates and evaluate which the radiosonde algorithm correlates well with the remote sensing observations. For example, in arctic climate (say Alaska) we would expect higher percentage of stable boundary layers and therefore the  $z_i$  estimates based on Richardson number might be well suited for training. We have clarified that the choice of reference data is indeed site specific in the updated manuscript (Page 8 Line 189 to 198).

With regards to the nighttime  $z_i$  estimates, this is a part of ongoing research by the larger community to define the true nocturnal boundary-layer height (Zilitinkevich and Baklanov 2002, Vickers and Mahrt 2004, Steeneveld et al. 2006, Richardson et al. 2013). During stable boundary layers, the determination of the PBL height is very uncertain. Turbulence in the stable boundary layer, can result from either buoyancy forcing or wind shear. At SGP C1, the nose of the low-level jet can also be used to define the height of the boundary layer (Sivaraman et al. 2013). Therefore, the subject of this article is not delve onto which nocturnal  $z_i$  estimates from the radiosondes are more accurate but rather to develop a methodology/framework to provide continuous nocturnal  $z_i$  estimates. We have made the point clearer in the updated manuscript that we are not doubting the results from the RF model but we are showing a methodology that can be adapted to future research needs associated with the depth of stable boundary layers (Page 8 Line 190 to 198). Moreover, in many stable boundary layer conditions, the true boundary layer height could be lower than the height of the first range gate of the lidar, so such a technique could provide accurate  $z_i$  estimates for such conditions.

During convective boundary layers, the definition is clearer and can more easily be discerned from radiosonde datasets. In Section 4, we provided a thorough evaluation during both convective and stable boundary layers compared to radiosonde estimates, which are assumed to be the true boundary layer height estimate at SGP C1. We hope from the results shown in Section 4 (Figure 5 and Table 3), readers can conclude that the Tucker method is not as accurate as the RF model  $z_i$  estimates.

**References:**

Zilitinkevich, S., & Baklanov, A. (2002). Calculation of the height of the stable boundary layer in practical applications. Boundary-Layer Meteorology, 105(3), 389-409.

Steeneveld, G. J., Van de Wiel, B. J. H., & Holtslag, A. A. M. (2007). Diagnostic equations for the stable boundary layer height: Evaluation and dimensional analysis. Journal of applied meteorology and climatology, 46(2), 212-225.

Vickers, D., & Mahrt, L. (2004). Evaluating formulations of stable boundary layer height. Journal of applied meteorology, 43(11), 1736-1749.

Richardson, H., Basu, S., & Holtslag, A. A. M. (2013). Improving stable boundary-layer height estimation using a stability-dependent critical bulk Richardson number. Boundary-layer meteorology, 148(1), 93-109.

Figure 10 and associated text: it is a bit confusing as to the whole purpose of this exercise. Just because the variance is being scaled by a higher PBL depth, the profile will look different. So not sure how it speaks to the Random Forest (RF) PBL depth being better than that derived by the Tucker method. Also, the variability of variance is probably huge, so the differences wouldn't be statistically significant anyways. This needs to be clarified in the text, or else removed from the manuscript. Thanks.

The last two sections in the manuscript are case studies showing the importance of an accurate boundarylayer height estimation. We agree that a higher boundary-layer height will change the profile, as we are scaling it with a different PBL height (Page 24 line 528). Since we have already shown that the RF model  $z_i$  is more accurate than the Tucker method, this section is showing the amount of uncertainty in using one method over the other at SGP C1. As shown in Page 24 Line 525, the average uncertainty in using the RF model  $z_i$  compared to the Tucker method  $z_i$  can result in approximately 10% uncertainty in convective velocity estimates and almost 15% in vertical velocity variance profiles. Normalized profiles are generally used in boundary-studies and atmospheric models (Lenschow et al. 1980), and understanding the importance of accurate  $z_i$  estimates is highlighted. We have provided additional motivation for this section in the updated manuscript (page 23 line 502 -507).

**Minor Concerns:**

Line 14: Might be better to say four years rather than multi-year. Thanks.

Agreed. Corrected.

Line 41: MISR is mis-spelled.

Since the abbreviation was only used once in the article, this has been removed in the updated manuscript.

Line 42:43: The satellites measure cloud top temperature from which the cloud top heights are calculated. During cloudy conditions, it is assumed that the PBL top corresponds to cloud top heights. This statement states that there has not been any validation of the satellite derived cloud top heights. Please add reference to support this, or else remove. Thanks.

For brevity, we have removed that statement from the paper. Although, we did some further research, and found only one recent paper (Böhm et al., 2019) which does a statistical evaluation of the MISR cloud base heights and ground-based Ceilometer observations.

Böhm, C., Sourdeval, O., Mülmenstädt, J., Quaas, J., & Crewell, S. (2019). Cloud base height retrieval from multi-angle satellite data. Atmospheric Measurement Techniques, 12(3), 1841-1860.

Line 58: you mean rely and not relay.

Typo corrected. Thanks.

Line 64: better word would be "lowest gate" rather than minimum range.

Agreed. Corrected in the updated manuscript.

Table 1: It will be good if you add units to the measurement features. Thanks.

Agreed, we currently have added units to the measurement features in Table 1.

Line 145-146: please revise this sentence. Thanks.

Agreed. We have added some additional text to this section, based on some of the earlier comments and comments from the other Reviewer. The sentence was rephrased as well. Please see Page 7 line 154 to line 164 for the changes.

Line 165: The numbers do not add up. Four years of data should equal 1460 days, not sure how you got 1785 days.

The ARM site has at least 2 radiosondes during daytime conditions and a higher frequency during some field campaigns, so for 4 years there are approximately 2920 radiosonde launches. We used 4 years of data and found approximately 1785 cases (not days) in our analysis. This typo was fixed and the statement has been rephrased to "…1785 cases with radiosonde data and daytime clear (identified as periods when surface heat flux is positive from sunrise to sunset and cloud base height is zero) or shallow cumulus conditions (identified as cloud base height less than 5 km from Doppler lidar and cloud fraction less than 0.1) for the years 2016 through 2019."

Line 285: you mean "hourly" cloud fraction greater than 0.1?

Yes, we use a rolling hourly average estimate of the cloud fraction to differentiate the measurements. The Doppler lidar, although, provides a cloud fraction estimate every 15 minutes. We have included this in the manuscript.

Figure 8: Please describe the vertical bars in the caption.

Yes, we have mentioned the vertical bars represent one standard deviation in the caption.

Figure 11: Looks like the LASSO simulations are able to accurately capture the development of the daytime PBL. I assume that the E3SM values are within the model range resolution as well. So, this is very good news for the modelling community and should be highlighted.

We thank the reviewer for the comment. We agree and have highlighted the motivation in the updated manuscript.